# A Confocal Microscopic Study of Gene Transfer into the Mesencephalic Tegmentum of Juvenile Chum Salmon, *Oncorhynchus keta*, Using Mouse Adeno-Associated Viral Vectors

**DOI:** 10.3390/ijms22115661

**Published:** 2021-05-26

**Authors:** Evgeniya V. Pushchina, Ilya A. Kapustyanov, Ekaterina V. Shamshurina, Anatoly A. Varaksin

**Affiliations:** A.V. Zhirmunsky National Scientific Center of Marine Biology, Far East Branch, Russian Academy of Sciences, 690041 Vladivostok, Russia; ilyaak9772@gmail.com (I.A.K.); eshamshurina@rambler.ru (E.V.S.); anvaraksin@mail.ru (A.A.V.)

**Keywords:** adeno-associated virus, calcium binding protein HuCD, *Oncorhynchus keta*, dorsal thalamus, posterior tuberal region, hypothalamus, pretectal nuclei, postcommissural region, tegmentum, mesencephalic reticular formation

## Abstract

To date, data on the presence of adenoviral receptors in fish are very limited. In the present work, we used mouse recombinant adeno-associated viral vectors (rAAV) with a calcium indicator of the latest generation GCaMP6m that are usually applied for the dorsal hippocampus of mice but were not previously used for gene delivery into fish brain. The aim of our work was to study the feasibility of transduction of rAAV in the mouse hippocampus into brain cells of juvenile chum salmon and subsequent determination of the phenotype of rAAV-labeled cells by confocal laser scanning microscopy (CLSM). Delivery of the gene in vivo was carried out by intracranial injection of a GCaMP6m-GFP-containing vector directly into the mesencephalic tegmentum region of juvenile (one-year-old) chum salmon, *Oncorhynchus keta*. AAV incorporation into brain cells of the juvenile chum salmon was assessed at 1 week after a single injection of the vector. AAV expression in various areas of the thalamus, pretectum, posterior-tuberal region, postcommissural region, medial and lateral regions of the tegmentum, and mesencephalic reticular formation of juvenile *O. keta* was evaluated using CLSM followed by immunohistochemical analysis of the localization of the neuron-specific calcium binding protein HuCD in combination with nuclear staining with DAPI. The results of the analysis showed partial colocalization of cells expressing GCaMP6m-GFP with red fluorescent HuCD protein. Thus, cells of the thalamus, posterior tuberal region, mesencephalic tegmentum, cells of the accessory visual system, mesencephalic reticular formation, hypothalamus, and postcommissural region of the mesencephalon of juvenile chum salmon expressing GCaMP6m-GFP were attributed to the neuron-specific line of chum salmon brain cells, which indicates the ability of hippocampal mammal rAAV to integrate into neurons of the central nervous system of fish with subsequent expression of viral proteins, which obviously indicates the neuronal expression of a mammalian adenoviral receptor homolog by juvenile chum salmon neurons.

## 1. Introduction

Study of the patterns of neural networks functioning in ontogenesis and their ability to incorporate new elements during life is an urgent issue of neurobiology. The successful implementation of such neurobiological research requires effective and accurate methods. In this sense, recombinant adeno-associated viral vectors are an effective tool that can be used to both target and manipulate certain subtypes of neurons (determined based on gene expression, location, and connections) and non-neuronal cells in the nervous system [1]. In recent years, clinical trials of the use of adeno-associated virus (AAV) vectors for the treatment of several genetic diseases, including central nervous system (CNS) disorders, have shown intriguing results [2]. Recombinant adeno-associated viruses (rAAV), derived from non-pathogenic and non-genotoxic parvoviruses that are extracted from their natural DNA genomes, have become not only a powerful tool for brain research but also one of the most promising therapeutic agents for the treatment of genetic diseases affecting the nervous system. A clinical trial of systemic rAAV delivery for the treatment of spinal muscular atrophy in infants has shown remarkable safety and therapeutic efficacy [3]. Despite the recent close attention to the potential of rAAVs and their use as biotherapeutic agents, there are still some problems with their practical application for the treatment of neurological diseases. Furthermore, there is a multitude of recently discovered and poorly studied factors that researchers should consider when developing vectors and their delivery methods for efficient transduction of CNS tissues. One of the major problems in the efficient delivery of genes to the target tissue of AAV is the pre-existing immunity to a wild-type AAV in the form of antibodies to the capsid of the wild-type virus. This is especially important for systemic administration of the vector, since the highest concentrations of anti-AAV antibodies are found in blood [4].

Currently, there are not many animal studies that conclusively demonstrate the use of AAV, unlike adenoviruses, on the central nervous system. Most of these studies were carried out on mice [5,6]. Studies on cats have evaluated the ability of three AAV serotypes (1, 2, and 5) to transduce cells in the brain. The lysosomal enzyme human β-glucuronidase was used as a reporter gene, since this gene differs in thermal stability. Vectors were injected into the cerebral cortex, caudate nucleus, thalamus, *corona radiata*, internal capsule, and *centrum semiovale* of 8-week-old cats [7]. The gene expression in the cat brains was assessed using in situ hybridization and enzyme histochemistry at 10 weeks post-surgery. The AAV2 vector was able to transduce gray matter cells, while the AAV1 vector resulted in a higher gray matter transduction than AAV2, as well as in white matter transduction. AAV5 did not induce appreciable transduction in the cat brain.

Until recently, adeno-associated virus 9 (AAV9) was considered an AAV serotype most effective for crossing the blood–brain barrier (BBB) and transducing CNS cells after systemic injection [8]. However, a recently developed capsid, AAV-PHP.B, crosses the BBB with even greater efficiency. In studies on C57BL/6 mice at 6 weeks of age, scAAV2/9-GFP or scAAV2/PHP.B-GFP was injected intravenously at equivalent doses [6]. Using a mouse model, it was investigated how much the expression of the CNS transgene can be increased using AAV-PHP.B, which carries the self-complementary (sc) genome. With the use of the scAAV2/PHP.B-GFP vector, the transgene expression in the brain and spinal cord showed more widespread CNS transduction and higher transgene expression levels at 3 weeks post-injection. In particular, an unprecedented level of astrocyte transduction in the cerebral cortex was revealed using the ubiquitous CBA promoter. In comparison, the neuronal transduction was much lower than previously reported. However, a strong expression in motor neurons of the spinal cord was observed using the human synapsin promoter [6]. To scale up the comparative studies on a rhesus monkey model, the transduction efficiency of AAV-PHP.B was evaluated [9]. Observations have shown a differential transduction pattern in macaques, with an extensive cortical and spinal cord transduction observed after intratectal administration and a very low transduction after intravascular administration. These results suggest that AAV-PHP.B may be a useful gene therapy vector for neurological disorders associated with the cerebral cortex or spinal cord.

Adenoviruses are known to be capable of infecting all classes of vertebrates [10] and are relatively weakly pathogenic for lower vertebrates. Thus, these agents can be very suitable for the transfer of genes into fish cells for vaccination or for other experimental studies where certain genes are expressed. Recently, there have been many technological advances in the development of AdV vectors including, in particular, the modulation of genetically modified tropism, which can significantly improve the infectivity of the base vector [11,12].

Over the past five decades, special attention has been paid to the study of mammalian and avian adenoviruses such as the mastadenovirus and aviadenovirus [13]. In the first molecular study of the AdV strain isolated from poikilothermic vertebrates, the complete genome of the leopard frog *Rana pipiens* isolate was analyzed [14]. The adenovirus of this frog (FrAdV-1) was found to be related to an unusual avian adenovirus, turkey adenovirus type 3 (TAdV-3), which is also referred to as hemorrhagic enteritis virus [15]. In recent studies, the AdV (SnAdV-1) genome isolated from the corn snake *Elaphe guttata* has been studied in sufficient detail [16]. The presence of AdV-like particles has been observed in several fish species; however, currently, only the gene isolate of the white sturgeon *Acipenser transmontanus* is available. Nevertheless, due to a significant discrepancy in gene sequences, this virus can be taxonomically isolated into a separate class of adenoviruses [17]. Initial studies based on phylogenetic analysis of the PCR-amplified part of the viral DNA polymerase gene confirmed its adenoviral origin and differences from other AdVs [18]. Additional information about the WSAdV-1 genome obtained a phylogenetic analysis of complete genes, and determination of the taxonomic position of WSAdV-1 in the Adenoviridae showed significant differences. Further characterization of this AdV in the future may lead to the development of new vectors for the delivery of genes that are perfectly suited for fish. Nevertheless, available data on the potential for an AdV-mediated gene delivery into fish cells in vitro or in vivo necessitates further study.

Viral vectors are widely used in the gene therapy, and also in clinical trials carried out every year for the treatment of various human pathologies using an increasing number of pharmacological agents. However, preclinical testing is time consuming and costly, especially during the developmental phase. Currently, fish are increasingly frequently used as alternative models for testing adenoviral vectors in vivo [19,20]. As a result of our preliminary studies of the brain regions in juvenile Pacific salmon, data were obtained on the ability of recombinant mammalian hippocampal AAV to infect cells of the cerebellum [21] and mesencephalic tegmentum [22]. Studies on *Danio rerio* embryos have shown an efficient transduction of adenoviral vectors E1/E3 with a single intracranial injection into fish embryos [19]. An essential characteristic is that highly productive AdV vectors allow the transgene to be stably expressed in the fish organism. It was found that the rainbow trout cell line CHSE-214 has a higher tropism for AdV than other cell lines of carp, white sturgeon, fat-headed gudgeon, sea bass, and tilapia, and it has a high homology of the Coxsackie adenovirus receptor required for AdV attachment [23,24]. The tropism for AdV in fish is likely to be species-specific.

In neurobiological studies, AdV vectors are used for various purposes such as the labeling of certain neurons and neuronal populations [25], tracing of a neuronal cell line [26], and modulation of neuronal function [27]. In this sense, the use of fish as neurogenic models in comparison with mammals has a number of advantages: in particular, the small size of adult fish requires smaller space for their cultivation, maintenance facilitates, and experimental manipulations, and it allows using more animals at a lower cost. Another advantage is that fish can reproduce in captivity: for example, one pair of *Danio rerio* can produce hundreds of eggs per week, which can be fertilized externally with subsequent embryonic development, which allows study of the early and subsequent stages of ontogeny. Embryonic and post-embryonic development of fish occurs relatively quickly, providing the opportunity to detect any abnormality within a few days. Furthermore, genetic abnormalities such as knockouts and knockdowns in *Danio rerio* are easy to obtain. About 70% of the genes coding for human proteins associated with diseases have equivalent genes in *Danio rerio*, which allows researchers to study human diseases on a large scale [28]. Thus, the use of viral vectors in fish contribute to further genetic analysis of neural functions and studies of neurogenesis in adult animals.

GCaMPs are genetically encoded indicators of calcium which contain a fluorophore and consist of green fluorescent protein (GFP) associated with calmodulin and the M13 peptide [29]. After binding calcium to the calmodulin-M13 system, conformational changes occur in the obtained protein complex, which leads to an increase in GFP fluorescence. GCaMP6 is one of the new genetically encoded indicators of calcium characterized by a higher signal-to-noise ratio and improved temporal kinetics compared to previous generations of calcium indicators [29]. There are many serotypes of adeno-associated viruses (AAV) available, each of which includes a separate viral capsid protein and provides different characteristics of transduction within the brain [30]. Some AAV serotypes are transported along the neuronal projections of the injected nucleus or brain region [31].

However, large-scale transfer of genetic material into fish cells, either for targeted gene expression or for DNA vaccines, is currently limited. The strategies evaluated to date are based on the use of a variety of physical methods, including injection [32], particle bombardment [33], chemical transfection reagents [34], and electroporation [35] into fish eggs. Transduction in fish muscle tissue is also achieved by the pistol bombardment of gene particles [23]. Little data on gene delivery into fish cells in culture were obtained using the vesicular stomatitis virus glycoprotein (VSVG)-pseudotyped retrovirus [36], and the Semliki forest virus [37]. However, efficient systems for in vivo delivery of genes into juvenile fish cells are still under development.

Previously, studies were initiated to evaluate the effectiveness of adenoviral vectors for the delivery of genes into fish cells, both in vitro and in vivo. These studies were based on vectors of human AdV serotype 5, which are used in clinical trials in humans, but they have not been evaluated for the delivery of genes into fish cells. Thus, to date, little is known about viral receptors in fish cells such as the Ad (Ad5Luc1) and Ad (Ad5LucRGD) adenoreceptors with natural tropism for the Coxsackie adenovirus receptor (CAR) and the adenovirus receptor. In these experiments, gene expression was detected in cell lines using both vectors [38]. The expression of Ad5LucRGD is much higher than Ad5Luc1 in most cell lines, with the exception of CHSE-214. The transduction of CHSE-214 cells with Ad5Luc1 can be blocked by increasing the competitive inhibitor; it is assumed that these cells have a homologue of CAR that binds AdV. In such studies, several methods have been used to deliver the overexpressed gene in vivo via intramuscular injection, although the transduction efficiency was comparatively low. Thus, it was found that some teleost fish cell lines are capable of adenoviral transduction, and one of the cell lines expressed the human serotype of a CAR homologue. In vivo studies have shown that rainbow trout tissue muscles can transduce AdV vectors, suggesting an alternative gene delivery strategy for this animal.

AAVs provide a rapid expression of transgenes in spatially defined regions of the brain and can target specific subsets of cells using specific promoters and cross-sectional genetic approaches. As a consequence, viral gene transfer has become an important tool for a wide range of applications, including optical measurements and the manipulation of neuronal activity using genetically encoded calcium indicators (GECI) and optogenetic probes, respectively [39,40]. However, according to [41], the commonly used AAV or lentiviruses cannot induce a detectable expression of transgenes in the zebrafish brain. Therefore, the fast, flexible, and economically efficient ways of expressing transgenes in zebrafish, which do not require long-term production of stable transgenic lines, are desirable.

Based on our previous studies, which investigated the high neurogenic potential of the brain of juvenile salmonids [42,43], we hypothesized that a high production of neurons in the juvenile salmonid brain may provide better rAAV transduction. Based on results of the preliminary studies [20,21,22], we suggested that a single injection of the transferred rAAV vector into the tegmental region of juvenile chum salmon, *O. keta*, would lead to a wide spread of the reporter gene, thereby covering a large area of the mesencephalic tegmentum (and/or other regions of brain), including various types of cells. The aim of this work was to study the feasibility of transfer of recombinant adenovirus (rAAV) from the hippocampus of CA1 mice to the brain cells of the juvenile chum salmon at 1 week after a single injection of the vector with the subsequent determination of the phenotype of rAAV-transgenic cells by CLSM.

## 2. Results

At 1 week after the injection of the rAAV into the mesencephalic tegmentum of the juvenile chum salmon, GFP-expressing cells were identified in various regions of the periventricular thalamus, hypothalamus, pretectal region, postcommissural region, basal mesencephalon, posterior tuberal region of the diencephalon, and areas immediately adjacent to the injection zone (Table 1). After 1 week, we identified a rather large post-injection cavity (IL) in the brain of juvenile chum salmon, which was detected at several levels of the tegmentum, surrounded by GFP+ granules, GFP-expressing nuclei, and cells that have a gradient distribution and form discrete populations located in different regions of the mesencephalon and diencephalon.

### 2.1. Anterior Hypothalamic Ventricle

Figure 1A–F presents Z-stacks, which show the results of scans of the brain sections and controls in different channels: DAPI staining of the brain sections, indicating the area containing GFP/HuCD-labeled cells (Figure 1A, pictogram). The rostralmost area of the brain containing transduced AAV cells was the area of the anterior hypothalamic ventricle associated with the caudal diencephalon (Figure 1A, pictogram). The morphometric parameters of AAV+, HuCD+, AAV+/HuCD+, and DAPI-stained nuclei of cells of the anterior hypothalamic ventricle are shown in Table 1. DAPI staining revealed numerous labeled cell nuclei and their small clusters (Figure 1A, in white dashed ovals). The largest accumulation of DAPI stained nuclei was found in the area adjacent to the medial part of the hypothalamic ventricle (Figure 1A, outlined by yellow rectangle). Scanning in the green channel revealed GFP+ elements varying in size (Figure 1B, Table 1). AAV hippocampal-specific mice were selected to evaluate the vector transduction and distribution of GFP+ proteins in cells, nuclei of cells, and other subcellular elements. Several clusters of GFP+ granules (populations №1 and 4), nuclei (population № 2), and cells (population № 3) were identified in the area of the anterior hypothalamic ventricle in the juvenile chum salmon (Figure 1B); morphometric parameters of the identified structures are shown in Table 1. The granules identified in the area of cell accumulation of the medial (MH, population № 1) and lateral hypothalamus (LH population № 4) are the smallest weakly and/or moderately immunofluorescent (IF) elements containing AdV GFP+ proteins (Table 1). Another population (№ 2, outlined by red square) contained small rounded or oval cells and nuclei in the area of the lateral hypothalamic ventricle (Figure 1B, Table 1). A small accumulation of GFP+ cells was found in the dorso-lateral hypothalamic region (population № 3), including the largest GFP+ cells in the anterior hypothalamic ventricle (Figure 1B, Table 1). IF was observed mainly in neuron bodies (Figure 1B). In the control brain sections, GCaMP6m-GFP was not labeled in the animals that received an injection of 0.1% PBS (Figure 1C). A neuron-specific intense/moderate IF signal of HuCD was detected in populations № 2–4 (Figure 1D). Intense HuCD+ immunofluorescence was detected in small rounded and oval neurons (Figure 1D, Table 1). In the control brain sections, no HuCD immunolabeling was detected in the animals with the lack of secondary antibodies (Figure 1E). The overlap of three transmission channels showed colocalization of the GFP signal and HuCD in populations 2 and 3 of neurons (Figure 1F) and the lack of colocalization in nuclei (population № 4) and granules (population № 1). Along with AAV+ cells, HuCD+ neurons lacking the transgenic signal were also identified (Figure 1F, green arrows). The results of ANOVA for the anterior hypothalamic ventricle are shown as the proportions of AAV+, HuCD+, and AAV+/HuCD+ cells in Figure 1G. Significant intergroup differences (# < 0.05) were found between the groups of AAV+ and HuCD+ cells and the groups of HuCD+ and AAV+/HuCD+ (## < 0.01) cells (Figure 1G).

### 2.2. Posterior Tuberculum Area

For a detailed characterization of the pattern of the AAV transgenic transfer into various cell types, we examined more in detail the posterior tuberculum area (PTA), which is the rostromedial boundary of the AAV distribution at 1 week after a single injection into the tegmentum of juvenile chum salmon. DAPI staining in PTA revealed morphologically heterogeneous nuclei forming typical dense clusters (outlined by white square), as well as resembling multilayered clusters (outlined by pink dotted oval), which were abundantly vascularized (orange arrows) (Figure 2A, Table 1). The study made it possible to identify in most cases the heterogeneous morphology of stained nuclei, often containing numerous nucleoli, which indicates a high intensity of cellular metabolism (Figure 2B, Table 2). We classified elongated nuclei, stained with DAPI, as belonging to migrating cells of another type (Figure 2B). The study of the GFP expression in PTA revealed two populations of cells with moderate intensity of IF (Figure 2C, Table 1). The accumulation of GFP+ cells in subventriculae zone (SVZ) with moderate and high signal intensity (population №1) had a heterogeneous composition (Figure 2C, inset). In this cell population, moderately and intensely GFP+-labeled neurons, intensely labeled nuclei, and moderately labeled granules were revealed (Figure 2C, inset, Table 1). The posterior tuberal nucleus (PTN) included a GFP+ heterogeneous cluster (population № 2) of neurons and nuclei (Figure 2D, Table 1). We evaluated the level of GFP expression in PTN as intense/moderate (Table 1). Numerous GFP+ granules of subcellular size and diffuse localization were identified around PTN (Figure 2D, Table 1). Moderate and intense IF-labeling of HuCD in PTA was detected in cells of populations № 1 and 2 (Figure 2F). Moderate IF labeling of HuCD dominated in PTN; single HuCD+ neurons were identified around PTN (Figure 2F, Table 1). Colocalization of GFP/HuCD was detected in intensely labeled neurons of population № 1 (Figure 2G, Table 1). A moderate level of GFP/HuCD colocalization signal was identified in neurons and nuclei of population № 2 (Figure 2H, Table 1). A weak GFP/HuCD colocalization signal was identified in granules surrounding PFT (Figure 2H, Table 1).

The ANOVA data for PTA are shown as the proportions of GCaMP6m-GFP+, HuCD+, and GCaMP6m-GFP+/HuCD+ cells in Figure 2I. Significant intergroup differences were found between the GCaMP6m-GFP+ and HuCD+ groups, as well as between GCaMP6m-GFP+ and GCaMP6m-GFP+/HuCD+ (# < 0.05) cells and the groups of HuCD+ and AAV+/HuCD+ cells (## < 0.01) (Figure 2I).

### 2.3. Dorsal Thalamus

After the AAV injection into the tegmentum, transduced cells and nuclei were found in the periventricular nuclei of the thalamus: dorsal (Dth) and ventral (Vth), pretectal nucleus (PRT) and rostral part of the reticular formation (rRF) (Figure 3A, pictogram). Staining with DAPI in Dth and Vth revealed clusters of small rounded nuclei, which formed periventricular cell arrays of increased distribution density (Figure 3A, Table 1). In the deeper parenchymal layers of the thalamus, we identified DAPI-stained fragments of microvessels (orange arrows) of the thalamic vascular plexus, elongated nuclei (pink arrows), sometimes forming heterogeneous complexes with oval cells (outlined by red rectangle), as well as local clusters of small oval nuclei (outlined by white dotted oval) forming neurogenic niches of parenchymal localization (Figure 3A). Dorso-laterally from Dth, DAPI-stained heterogeneous nuclei clusters were localized, forming a complex of pretectal nuclei (PRT) (Figure 3B, Table 1). Ventrally of PRT, extended arrays of nuclei of heterogeneous morphological composition and distribution density (outlined by white dotted square) were identified, forming thalamic parts of the reticular formation (rRF) (Figure 3B, Table 1).

The study of GFP expression in the periventricular thalamus revealed a dense aggregation of moderately GFP+ cells and intensely labeled granules in Dth (population № 1) (Figure 3C, inset, outlined by yellow dotted oval, Table 1). In Vth, less densely aggregated, weakly IF GFP+-labeled cells, surrounded by diffuse clusters of moderately IF GFP+-labeled granules, were found (Figure 3C, inset, outlined by orange dotted oval, Table 1). Single weakly IF GFP+-labeled cells were found in PRT, along with intensely IF GFP+-labeled cells and nuclei (population № 3 in Figure 3C, outlined by white dotted oval). GFP expression in rRF revealed a large heterogeneous population of intensely fluorescent cells, nuclei, and granules, located diffusely, at a significant distance from each other (Figure 3D, outlined by white dashed rectangle, Table 1).

The study of immunofluorescence of HuCD in the periventricular nuclei of the thalamus revealed moderate IF HuCD+-labeled oval cells in Dth (Figure 3F, inset, outlined by yellow dotted oval, Table 1) and moderate IF HuCD+-labeled rounded cells in Vth (Figure 3F, inset, outlined by orange, dotted oval Table 1). In Dth and Vth, few intensely IF HuCD+-labeled granules were revealed (Figure 3F). In PRT, small rounded intensely IF HuCD+-labeled cells and larger moderately IF HuCD+-labeled neurons were found (Figure 3F, outlined by white rectangle, Table 1). In rRF, we found intensely IF HuCD+-labeled cells of medium size and rounded in shape (Figure 3F, green arrows, Table 1), weakly IF HuCD+-labeled cells of the same size (white arrow), and large weakly IF HuCD+-labeled neurons (Figure 3F, yellow arrows, Table 1). The overlap of three transmission channels (which showed colocalization of the DAPI, GFP, and HuCD) in the region of the periventricular nuclei of the thalamus indicated moderate/low IF-labeling in the case of colocalization of GFP and HuCD in neurons of Dth and Vth (populations № 1 and 2) (Figure 3G, Table 1). However, single GFP–/HuCD+ cells were found in Vth (Figure 3G, orange dotted rectangle). In PRT, after Z-stacks overlapping, the most complex pattern of colocalization of DAPI, GFP, and HuCD was revealed, in which intense and moderate IF and colocalization of GFP and HuCD were detected in nuclei and oval neurons of medium size (Figure 3G, inset, outlined by white rectangle, Table 1). Large polygonal HuCD+/DAPI+/GFP– neurons that formed a cluster in PRT were adjacent, on one side, to a population of intensely IF GFP+/HuCD+-labeled cells and nuclei (Figure 3G, white dotted rectangle) and, on the other side, to the vascular sinus with migration patterns (see Figure 3G, inset, outlined by white rectangle). In PRT, a population of large differentiated HuCD+/DAPI+/GFP– neurons, lacking GFP expression, was localized throughout the nucleus (Figure 3G, green arrows) and formed a complex pattern of neuronal distribution (Figure 3G, outlined by white dashed rectangle). Similar differentiated neurons were found in the ventral region within rRF (Figure 3G, Table 1). Along with large GFP–/HuCD+ cells, rRF showed patterns of colocalization of intensely IF GFP+/HuCD+-labeled cells and nuclei (Figure 3H, outlined by white dotted square) and separate intensely IF-labeled granules.

The results of ANOVA for Dth, Vth, PRT, and rRF are shown in Figure 3I. Significant intergroup differences (# < 0.05) were found between the groups of GCaMP6m-GFP+ and GCaMP6m-GFP+/HuCD+ cells and the groups of HuCD+ and GCaMP6m-GFP+/HuCD+ cells (Figure 3I).

### 2.4. Postcommissural Area

Other clusters of labeled cells were found near *comissura transversa* located in the rostral part of the basal mesencephalon (Figure 4A, pictogram), where several regions containing GFP labeling were revealed. The DAPI staining revealed dense nuclei clusters in the periventricular zone (PVZ) as part of the posterior tuberal nuclei (PTN) adjacent to the lumen of the hypothalamic (III) ventricle (Figure 4A). The deeper layers of the brain were extensively vascularized and included cell aggregations partially containing cells from PTA and the rostral part of the reticular formation (Figure 4A). The study of the distribution of GFP-expressing cells in the postcommissural area showed the presence of four unequal populations of cells and nuclei (Figure 4B, Table 1). The dorso-medial posterior tuberal group (population №1) contained moderately GFP+-labeled granules in the PVZ and weakly IF-labeled cells in the subventricular zone (SVZ) (Figure 4B, Table 1). Another periventricular cluster (population № 2) contained few GFP+ granules similar to those in population № 1 (Figure 4B). Within mesencephalic reticular formation (MRF), there were intensely IF GFP+-labeled single cells forming population № 3 (Figure 4B, Table 1). The largest accumulation of intensely IF GFP+-labeled nuclei and less intensely labeled GFP+ cells (population № 4) was found in the structure of MRF (Figure 4B, Table 1). Weak IF HuCD labeling was found in populations № 1 and 4 (Figure 4C, Table 1). Intense IF HuCD+ labeling was detected in small oval-shaped neurons in population № 3 and in smaller nuclei of population № 4 (Figure 4C, Table 1). Colocalization of GFP/HuCD was found in intensely labeled neurons in population № 3 (Figure 4D, Table 1). A moderate level of GCaMP6m-GFP+/HuCD colocalization signal was identified in nuclei of population № 4 (Figure 4D). A weak GCaMP6m-GFP+/HuCD colocalization signal was identified in PVZ granules (population № 2) and SVZ neurons (population № 1).

The results of ANOVA for the postcommissural area is shown as the proportions of GCaMP6m-GFP+, HuCD+, and GCaMP6m-GFP+/HuCD+ cells in Figure 4E. Significant intergroup differences (# < 0.05) were found between the groups of GCaMP6m-GFP+ and HuCD+ cells, groups of GCaMP6m-GFP+ and GCaMP6m-GFP+/HuCD+ cells, and groups of HuCD+ and GCaMP6m-GFP+/HuCD+ cells.

### 2.5. Dorso-Medial Tegmentum

In the dorso-medial tegmentum (DMT), DAPI staining revealed a typical pattern of distribution of stained nuclei, sometimes forming small clusters (Figure 5A), a sparse distribution of microvessels compared to that in the hypothalamic region, and occasional patterns of migration of cells with elongated nuclear morphology (Figure 5A, Table 1). The periventricular zone of tegmentum in juvenile chum salmon has a large number of immature (blast) cell forms characterized by high nuclear-cytoplasmic ratios, large nuclei, and a narrow rim of the cytoplasm (Figure 5A). The results of scanning along the green channel showed that AAVs in the region of DMT were embedded in neurons and/or nuclei of various types, forming three populations (Figure 5B, Table 1). In population № 1, a weakly IF GFP-labeled protein was observed in small neurons and nuclei (Figure 5B, Table 1). A larger aggregation of intensely fluorescent nuclei and granules was found in population № 2 (Figure 5B, Table 1). The largest, diffuse aggregation of nuclei and cells with intense IF GFP labeling formed population № 3 (Figure 5B, Table 1). Morphometric parameters (M ± SD) of nuclei and cells of various areas of the midbrain, in which GCaMP6m-GFP expression was detected, are shown in Table 1. The results of scanning through the red channel showed the presence of moderately and weakly IF HuCD+-labeled cells in population № 1, intensely labeled small HuCD+ neurons in populations № 2 and № 3, and weakly labeled HuCD+ large neurons in population № 3 (Figure 5C, Table 1). The study of the overlapping of Z-stacks of DAPI, GFP, and HuCD labeling of the DMT region showed the colocalization of GFP and HuCD in 18.1% of cells (Figure 5D, Table 2). In population № 1, a few small, weakly fluorescent cells with GFP/HuCD colocalization were identified (Figure 5D, Table 1). In populations № 2 and № 3, colocalization was found in intensely labeled small cells but not in granule-like particles (Figure 5D, Table 1). In population № 3, colocalization was absent in large, weakly fluorescent HuCD+ neurons (Figure 5D, Table 1). The results of the ANOVA test in DMT are shown in Figure 4E. Significant intergroup differences (# < 0.05) were found between the groups of AAV+ and HuCD+ cells and between the groups of HuCD+ and AAV+/HuCD+ cells (Figure 5E).

### 2.6. Dorso-Lateral Tegmentum

In dorso-lateral tegmentum (DLT), DAPI staining revealed typical patterns of nuclear distribution, among which there was a dilated circulatory sinus (orange arrows) and a post-injection cavity (IL) filled by numerous heterogeneous stained nuclei and cells (in the white square, Figure 6A).

When scanning along the green channel in the DLT, three populations of GFP+ cells and nuclei were identified, which are surrounded by small GFP+ granules (Figure 6B, Table 1). The largest number of intensely IF GFP+-labeled cells surrounded by GFP+ granules was found in population № 1 (Figure 6B, inset, Table 1). In the ventro-lateral direction from population № 1, GFP+ nuclei were found, which were also present in population № 2 (Figure 6B, Table 1). In contrast to population № 1, GFP+ cells were not identified in population № 2, but there were few and sparse moderately IF-labeled granules (Figure 6B). In population № 3, partially located in the area of IL, single moderately IF GFP+-labeled nuclei and granules were identified (Figure 6B, Table 1). Intense IF HuCD-labeling in DLT was found in cells of population № 1 (Figure 6C, inset, Table 1). Intensely IF-labeled small cells and moderately IF GFP+-labeled granules were detected in populations № 2 and № 3 (Figure 6C, Table 1). Colocalization of GFP/HuCD was detected in intensely IF-labeled neurons of population № 1 (Figure 6D, Table 1). Moderate IF-labeling and colocalization of GFP/HuCD were identified in the nuclei of population № 2 (Figure 6D, Table 1). Weak IF-labeling and colocalization of GFP/HuCD were identified in the GFP+ granules surrounding the neurons of population № 1 (Figure 6D, Table 1).

The results of the ANOVA test in DLT are shown in Figure 6E; significant intergroup differences were found between the groups of GCaMP6m-GFP+ and HuCD+ cells, between the groups of GCaMP6m-GFP + and GCaMP6m-GFP+/HuCD+ cells (# < 0.05), and between the groups of HuCD+ and GCaMP6m-GFP+/HuCD+ cells (## < 0.01).

### 2.7. Edinger–Westphal Nucleus

After the AAV injection into the tegmentum of juvenile chum salmon, transduced cells were detected in the small cell part of the nuclei of the oculomotor complex: the Edinger–Westphal nucleus (EWN), or the accessory nucleus of the third nerve, *nucl. accessorius nucleus oculomotorii*. This nucleus was close to the post-injection cavity (IL) (Figure 7A) and was formed by small cells located near the main nucleus of the oculomotor nerve, being the source of autonomic fibers. DAPI staining showed single clusters of nuclei/cells forming neurogenic niches (outlined by white oval) and numerous oval or elongated nuclei with single nucleoli (Figure 7A, Table 1). In the area of the additional optical system, an abundant capillary network was revealed along the elongated endothelial nuclei (orange arrows) and fibers of the oculomotor nerve (pink arrow). The EWN region was morphologically heterogeneous with a high cell density and distribution (Figure 7B, Table 1).

As a result of the study of GFP expression in EWN, a gradient distribution of the labeled elements was revealed (Figure 7C). A dense accumulation of GFP+ granules was detected in the immediate vicinity of IL (Figure 7C, outlined by white oval). A large population of intensely IF GFP+-labeled cells (Figure 7C, Table 1) surrounded by diffuse, small, intensely IF-labeled granules (Figure 7C) was ventrally adjacent to IL (Figure 7C, outlined by dashed rectangle). GFP+ cells in EWN were distinguished by a heterogeneous IF, which was significantly higher than that of granules (Figure 7D). The study of IF HuCD showed intense labeling of neurons with EWN and moderate IF labeling of granules (Figure 7E). A detailed examination of EWN revealed heterogeneous HuCD+ neurons of round and oval shape, which were small and medium in size (Figure 7F, Table 1). The study of the overlapping of Z-stacks of DAPI, GFP, and HuCD labeling of the EWN region showed a high degree of colocalization of GFP and HuCD in neurons and a lower degree in granules (Figure 7G); nevertheless, single GFP–/HuCD+ neurons were identified in EWN (Figure 7G). The intensity of the GFP and HuCD colocalization signal in neurons was high or moderate; in granules, it was moderate/weak (Figure 7H).

ANOVA data in EWN are shown as the proportions of AAV+, HuCD+, and GCaMP6m-GFP+/HuCD+ cells in Figure 7I. Significant intergroup differences (# < 0.05) were found between the groups of GCaMP6m-GFP+ and GCaMP6m-GFP+/HuCD+ cells and between the groups of HuCD+ and GCaMP6m-GFP+/HuCD+ cells (Figure 7I).

### 2.8. Mesencephalic Reticular Formation

The mesencephalic reticular formation (MRF) was the caudal zone of adenoviral transduction at 1 week after the AAV injection into the tegmentum of juvenile chum salmon. A diagram of the area of AAV injection (shown by a red line), and the distribution of GFP-expressing cells in the brain of juvenile chum salmon in MRF is shown in the pictogram (Figure 8A). DAPI staining revealed numerous labeled cell nuclei and their small clusters (Figure 8A, Table 1). Along the injection lumen (IL), we revealed invaginations of the parenchymal tissue of the brain, with heterogeneous aggregations of stained nuclei localized at the base and forming reactive neurogenic zones (Figure 8A, inset). Similar local neurogenic niches were found in deep layers of the parenchyma of the basal mesencephalon within MRF (Figure 8A, outlined by white dotted ovals). In the MRF region, dorsal and ventral aggregations of intensely IF GFP+-labeled small cells (populations № 1 and № 2) and nuclei (populations № 2) were identified (Figure 8B); morphometric parameters of the identified structures are shown in Table 1. The granules found within the dorsal cells population of MRF (population № 1) and outside it in the SVZ and parenchymal zone (PZ) had moderate/weak fluorescence (Figure 8B, inset, outlined by red rectangle, Table 1). Intense IF HuCD labeling was detected in the dorsal (population № 1) and ventral (population № 2) clusters of small rounded and oval-shaped neurons (Figure 8C, Table 1). The overlapping of three transmission channels showed partial colocalization of the GFP signal and HuCD in the dorsal population № 1 of small neurons (Table 1); some of the intensely IF-labeled cells were HuCD+/GFP– (Figure 8D, inset, outlined by red rectangle). In the ventral population, colocalization of GFP and HuCD was more typical in small, moderately IF-labeled neurons (population № 2, inset, outlined by yellow rectangle, Table 1). Along with AAV+ cells, HuCD+ neurons lacking the transgenic signal were identified in the ventral population (Figure 8D, green arrows).

The results of ANOVA in MRF are shown as the proportions of GCaMP6m-GFP+, HuCD+, and GCaMP6m-GFP+/HuCD+ cells in Figure 8E. Significant intergroup differences (# < 0.05) were found between the groups f HuCD+ and GCaMP6m-GFP+/HuCD+ cells.

## 3. Discussion

Viral vectors are of great clinical interest due to their high efficiency, especially when the genetic material is intended to reach the nucleus [44]. Adenovirus was one of the first viruses to be adapted as a gene therapy vector [45]. These viruses without envelopes, having icosahedral capsids up to 100 nm in size and linear double-stranded DNA genomes of about 36 bp, provide high genetic stability and efficiency of integration into various cell types [46]. Once inside cells, viral capsids decompose in a programmed manner and guide the genome into the nucleus, where they remain in an episomal state.

### 3.1. Cytotoxicity

Early versions of the first generation E1/E3 vectors are a useful tool for in vitro genetic transmission, but the in vivo use is limited by the short duration of transgene expression [46]. This is mainly due to the fact that the residual expression of viral genes in transduced cells induces cytotoxic immune responses. In contrast, third-generation adenoviral vectors, also known as high-capacity vectors (HC-AdV), lack all viral coding genes, retaining only two cis-acting sequences: inverted terminal repeat (ITRs) and packaging signals (ψ). With their enhanced cloning ability, these vectors can maintain transgene expression in vivo for a long time in organs with slow cell cycle such as the liver or brain [47,48].

The study of transduction of the first-generation vectors and HC-Ad of the luciferase reporter gene in *Danio rerio* embryos showed that the E1/E3 and HC-Ad vectors (Ad-EGFP and HCA-EGFP, respectively) were designed with the same promoter and capsid, and, thus, equivalent levels of expression of these genes should be expected [19]. Despite the similarity of genetic tropism, the results of these experiments showed differences in the number of transduced individuals. This variability was due to different concentrations for each vector. The microinjector used to deliver the vector in these studies provided a fixed pressure rather than a fixed volume. In addition, it should be taken into account that the intensity of GFP expression in transduced cells could affect the vector genome. The strong viral enhancers present in the E1/E3 vectors can enhance transgene expression in contrast to the HC-Ad vectors [47]. Stronger expression in cells infected by Ad-EGFP can facilitate visualization under a fluorescence microscope, facilitating the perception at a higher transduction rate. The results of the study of the mortality of *Danio rerio* did not reveal any differences between different types of vectors applied, with the survival rate of fish being more than 80% [19]. In these observations, acute toxicity resulting from infection with the introduction of viral genes was excluded.

Previous studies on juvenile chum salmon with a single injection of AAV also did not reveal toxic effects that reduce animal survivability [20,21,22]. In our experiments, we for the first time used GCaMP6 vectors with genetically encoded calcium indicatiors, which had not previously been used for fish. After a single injection of AAV into the tegmental parts of the brain of juvenile chum salmon, no deaths were recorded. Thus, the use of GCaMP6 vectors on juvenile salmonids makes it possible to visualize the region of GFP expression and calcium activity in brain cells, which gives us the opportunity to use these vectors for in vivo registration of the calcium signalization in AAV-transduced cells. Additionally, GCaMP6 vectors provide high signal intensity and clarity that does not degrade noticeably with dilution, ensuring high-definition imaging. GCaMP6 is one of the new genetically encoded indicators of calcium and is characterized by a higher signal-to-noise ratio and improved temporal kinetics compared to previous generations of calcium indicators [29]. The use of such calcium indicators is promising, since they have low cytotoxicity and are well tolerated by juvenile chum salmon.

According to data on *Danio rerio*, E1/E3-deleted vectors retained most of the viral genes in the genome, in contrast to HC-Ads [19]. Although gene transcription was activated by E1A, residual expression in mammalian cells has been described in the absence of the E1 region. Therefore, under experimental conditions, cells transduced with Ad-EGFP can express adenoviral proteins and also more GFP, as compared to HCA-EGFP. Both elements can compromise the viability of the transduced cells. Although they did not have a noticeable effect on the early development of *Danio rerio* embryos, the situation changed markedly in the long-term survival study [1]. A marked increase in mortality in *Danio rerio* occurred at 72 h post-transduction. This was due to environmental changes and the fact that *Danio rerio* have a high reproductive rate, which suggests that, individually, the embryo survival rate would decrease [19]. Instead, all *Danio rerio* treated with Ad-EGFP died within 11 days post-injection and showed more apathetic behavior with slow movements, which made them easier to catch. However, more than 30% of HCA-EGFP-treated fish showed normal behavior when observed for 35 days. The probable causes of these differences have been identified. It is assumed that the high level of GFP expression in cells infected with Ad-EGFP caused accumulative toxicity in the cells, which was not evident during the first 72 h.

Theoretically, the involvement of a cellular immune response against GFP or viral epitopes expressed by cells is unlikely, taking into account that the injection was done to juvenile chum salmon, which still have an immature immune system [49]. However, it is possible that the immune tolerance was not fully established under assay conditions and that the persistence of the antigenic expression triggered later immune responses [50]. In this regard, the main difference between cells infected with HCA-EGFP, Ad-EGFP, and GCaMP6-GFP was the presence of adenoviral antigens in the latter, which have been described as an excellent target for cytotoxic reactions in mammals [51]. Thus, cumulative cellular toxicity caused by viral proteins and/or high GFP expression is the most likely explanation for the decreased survival rate of zebrafish infected with AdV-EGFP, which was not detected in juvenile chum salmon, although immune responses against residual viral gene expression are not excluded.

Additionally, maintenance of transgene expression in *Danio rerio* by the HC-AdV vector was established by fluorescence monitoring for up to 32 days [19]. Thus, for clinical use, repeated administrations should be within approximately this time period, suggesting similar rates of cell replication in the region of interest. According to our observations, long-term expression of the transgene (more than 100 days) in 2-year-old chum salmon juveniles also does not cause toxic effects with a sufficiently clear signal and the lack of the effect of transgene dilution (unpublished data). However, other studies on rodents, along with other mammals, have shown a long-term expression, over one year [48]. This may be due to the higher cellular replication rate in embryos compared to that in adults, which accelerates the dilution of transgenes [47]. In fact, the loss of expression corresponds to the second half of the embryonic transition from larva to juvenile stage (from three up to about 30 days post-fertilization), which is a period when they also increased significantly in size, and their morphology supports this explanation.

### 3.2. Receptor Specificity

One of the most noteworthy results was the discovery of a functional homologue of Coxsackie adenoreceptor (CAR) in fish cells, primarily on the CHSE-214 line. To date, human and mouse CARs (hCAR and mCAR, respectively) have been cloned and characterized [52]. CAR belongs to the immunoglobulin (Ig) receptor superfamily and has two extracellular Ig-like domains. The normal function of CAR is currently unknown. Efforts to develop CAR knockout in mice have not been successful so far, as CAR is suggested to perform some critical regulatory functions in embryogenesis. There is evidence that the loss of CAR in tumor cells is associated with the formation of a more aggressive malignant phenotype [53]. If the existence of CAR-like molecules in fish is confirmed in further studies, this may indicate that some critical functions require evolutionary conservation.

Adenoviral vectors used in human clinical trials contain a naturally occurring receptor-binding domain that directs the attachment of the virus to CAR on the cell surface. After primary binding, the virion is further mediated through interactions between RGD motifs in the protein pentone at the base of the fiber and cellular integrins [54]. In the absence of CAR, these viruses are assumed to use the RGD–integrin interaction as a primary binding agent, although less efficiently than with CAR [55]. Studies using various established human cell lines, primary lines, and human tumor material have shown that certain tissues are refractory for Ad infection due to CAR deficiency [56]. As integrins are more widely expressed, the transduction efficiency of many of these cells has been improved using tropism-modified vectors, in which the integrin-binding RGD motif is genetically incorporated into the highly available high loop cycle of the handle domain region [57]. These modified vectors are expected to be used in human clinical trials very soon. At present, little is known about the expression of CAR in fish cells, but it is assumed that in the case of the presence of CAR or its homologue on some cell lines, a wider expression of integrins is quite probable. In vitro results confirmed this hypothesis because most lines were more easily infected with RGD-modified vectors; however, there was no significant improvement in expression with the RGD vector in vivo [56]. This lack of in vivo efficacy needs to be clarified. Considering the above and the results of analyses based on the detection of gene expression, the activity of the cytomegalovirus (CMV) promoter requires further study. However, in vitro results indicate that the promoter was active in fish cells. Previously, it was shown that CMV-containing vectors were expressed in vivo in fish by intramuscular injection of plasmids and/or using a gene cannon [23,58]. Temperature may be another factor limiting the activity of the CMV promoter. Although attempts were made to address this problem, some studies have obtained negative results due to the poor viability of cell lines at temperatures required to maintain the viability of live fish. The study of the Semliki forest virus for the delivery of genes into fish cells in culture showed less efficiency at low temperatures [37]. The recently discovered AdV of white sturgeon may become a more effective vector for these purposes.

### 3.3. Transduction of AdV into Fish Cells

The transfer of AdV vectors can be useful for specific expression in muscles, if such delivery has the advantages of expression in muscles or in vaccine development [46]. However, the results obtained on the white sturgeon *Acipenser transmontanus* indicate that genes can be delivered and expressed using an AdV into fish cells under conditions suitable for human infection, at 37 °C. It was found that AdV species recently isolated using fish cells do not cause any pathogenesis in juvenile white sturgeon, which makes it possible to relatively easily observe the development of the human AdV in fish cells. This allows for the manipulation of viral binding or reengineering of the natural strategy for mass AdV immunization of fish. Thus, it is possible to develop gene transduction by immersion or some similar methods. Research results indicate the opportunity of using AdV for the transduction of teleost fish cells under culture conditions, using direct injections into fish muscle, as well as intracerebral injections [17,19,59]. This method of genetic transduction is functional; however, several problems regarding the modification of the virus need to be addressed for the method to become a practical means of delivery and expression of genes in live fish.

Microscopic studies showed the presence of adenovirus-like particles in the epidermal hyperplasia of cod [60] and limanda [61], and also in Japanese red sea bream with lymphocytic leukemia [62], but WSAdV-1 can still only be used as a fish adenovirus isolate [63]. The characterization of AdV in fish is of particular interest because it contributes to the further research into the adenovirus evolution. Recently, a hypothesis was formulated [64] that four genera of adenoviruses (mastadenovirus, aviadenovirus, atadenovirus, and siadenovirus) correspond to viral lineages that developed along with the four main classes of vertebrates (mammals, birds, reptiles, and amphibians). According to this assumption, fish adenoviruses differ from all AdVs studied so far.

In the past two decades, numerous works have been carried out to study the development of teleosts. In the zebrafish *Danio rerio*, mutagenesis was induced with various chemical agents; medaka *Oryzias latipes* was used for a relatively long time in genetic research, during which many spontaneous and induced mutations were found [59]. Interesting data were obtained by the cultivation of male germ cells and the nuclear transplantation into both zebrafish and medaka cells [65]. The results of these studies have shown the opportunity to transmit genetic changes to the germ line of cells. The finding of a gene homologous to CAR, which has 73.1% identity of the mouse CAR, in zebrafish but not detected in medaka, gave important information. It was found that the rainbow trout cell line CHSE-214 has a higher tropism for AdVs than other cell lines of carp, white sturgeon, bighead gudgeon, sea bass, and tilapia. It also has a high CAR homology, which is necessary for AdV attachment. The tropism for AdVs is probably species-specific. Although the medaka receptor that binds AdV is unknown, the lower efficiency of infection of medaka by the AdV may be due to the low affinity for mammalian AdV and/or the low level of receptor expression. The advantage of using such a methodology in the future is obvious; however, with the limitations of systems that could successfully and efficiently introduce foreign, capable of recombining with endogenous homologous DNA sequences into model organisms. Such methods are still under development. Some of the viral vectors, in particular adenoviruses, are quite suitable for such purposes, although they have not previously been available for the use in fish models.

### 3.4. Efficiency of Infection the Recombinant Mouse Hippocampal AAV on the Juvenile Chum Salmon Brain Cells

Recent studies have shown that the efficiency of the use of AdVs for incorporation into the fish genome varies greatly. In particular, the AdV infection of zebrafish cells was found to be a hundredfold more efficient than that of medaka cells [59,65]. However, according to [41], the commonly used AAV or lentiviruses cannot induce detectable expression of transgenes in the zebrafish brain. Therefore, the fast, flexible, and cost-effective ways of expressing transgenes in zebrafish, which do not require long-term production of stable transgenic lines, are desirable. The results of this study showed that the AAVs of the mouse hippocampus are efficiently expressed in various cells of the diencephalon and mesencephalon of juvenile chum salmon, *O. keta*. In our study, within 1 week after a single injection of the AAV into the mesencephalic tegmentum of juvenile chum salmon, transduced cells, nuclei, and GFP+ granules were detected rostrally and caudally of the injection site, which is probably associated with the ability of AAV to transport to the brain in anterograde/retrograde directions [31]. The data obtained indicate the ability of AAVs to be efficiently transduced into juvenile chum salmon brain cells, which confirms our preliminary data [21,22].

Extracellular vesicles (EV) are nanometer to micrometer-sized endogenous lipid particles released from cells. They carry many types of nucleic acids and proteins from the host cell, as well as many receptors on their surface [66,67]. Studies of the distribution of AAV in vivo have shown that EVs are a form of intercellular interaction and spread that allows them to quickly enter recipient cells [68]. Studies on the hepatitis A [69] and hepatitis C [70] viruses have shown that EVs can be used to evade antibodies to the virus from patients. In addition, EVs are now developed as therapeutic means for the delivery of nucleic acids and drugs [71]. The most interesting results of studies using AAV showed that EV-associated AAV (ev-AAV) may have unique properties of standard AAVs, which may be useful for in vivo gene therapy, including the evasion of antibodies.

Neural members of the Hu family of RNA-binding proteins such as, in particular, HuC and HuD play an important role in the differentiation and plasticity of neurons. Several in vivo and in vitro experiments indicate an important role of neuron-specific Hu proteins in the neuronal differentiation [72,73,74] and neuronal development in both the central and peripheral nervous systems [75]. The major roles of Hu proteins are due to their molecular functions, which affect a large number of target genes. Hu proteins influence many post-transcriptional aspects of RNA metabolism, from splicing to translation [76]. Recently, the specific nuclear functions of these proteins have been identified to determine a wide range of regulatory influences of the HuC and HuD proteins [77]. In addition, in recent years, increasingly more information has been published about the post-transcriptional regulation, which is important for ensuring the strict and precise control of gene expression. It is now widely accepted that every step of post-transcriptional processes, including splicing, polyadenylation, editing, nuclear export, RNA localization, RNA degradation, and translation, can serve as a regulatory point for modulating the protein production of a particular gene. RNA binding proteins (RBPs) play an important role in the maturation and function of mRNA. The Hu proteins are a group of classic RBPs, neurons are highly enriched in the HuC and HuD proteins. The embryonic lethal visual anomaly (ELAV) protein, which is a homologue of these proteins in *Drosophila*, is expressed exclusively in neurons and plays an important role in flies [7]. The HuCD proteins modulate many aspects of post-transcriptional regulatory events. Understanding the functions of these proteins on the molecular level will provide important information about the complex and coordinated biological activity of the nervous system.

A comparative study of the brain regions in which transgenic cells, nuclei, and granules were identified after the AAV injection showed different distribution densities in the diencephalon and mesencephalon. Table 2 summarizes the percentages of GCaMP6m-GFP+, HuCD+, and GCaMP6m-GFP+/HuCD+ cells in all the brain regions where GFP and HuCD expressions were detected. A summary diagram showing the comparative distribution of GCaMP6m-GFP+ and GCaMP6m-GFP+/HuCD+ in all brain regions where transgenes were detected at 1 week post-injection are shown in Figure 9. In the diencephalon, the maximum number of GCaMP6m-GFP+ and GCaMP6m-GFP+/HuCD+ cells at 1 week post-injection was found in the posterior-tuberal area, and a slightly lower index of amount of transgenic cells was found in the anterior hypothalamic ventricle (Figure 9). In the mesencephalic tegmentum, the respective indices were significantly lower; nevertheless, the maximum number of transgenes were found in the dorsal medial tegmentum (Figure 9).

In our study, different degrees of colocalization with the molecular marker of early neuronal differentiation HuCD were found in several brain regions containing GFP+ expression, which is an important finding on juvenile salmonids made for the first time. The recorded differences in the proportion of GFP+ cells and also in the colocalization of GFP+/HuCD+ show that AAV is capable of incorporating AAV in different areas of cells of early neuronal differentiation. We believe that these differences can be determined by both probabilistic reasons (random propagation of the AAVs) and quite predictable parameters of cell density (the highest in the DLT area and the minimum in the MRF area). In our studies, the relationship between the distribution of the AAV and the injection cavity was established only for MRF, EWN, and DLT. However, the maximum and minimum numbers of GFP+ cells were found in MRF and DLT, respectively (Figure 9). Thus, proximity to the injection zone does not specifically affect the number of transduced cells, while the density of cell distribution correlates with the number of GFP+ cells.

The use of CLSM evidently demonstrates that GFP-expressing cells after a single AAVs transduction are widely distributed and can be colocalized with the HuCD neuronal protein. The results obtained indicate that the GFP-expressing cells of the tegmentum of juvenile chum salmon have a proneuronal phenotype. However, the revealed differences in the colocalization of GFP/HuCD between different areas of the tegmentum of juvenile chum salmon can be explained by other factors such as, in particular, by the mode of confocal imaging of labeled cells, the conditions for immunofluorescent labeling, different tissue permeability for antibodies, etc.

The distribution of adenoviruses in the brain is often associated with the intracellular transport in antero- and retrograde directions. A comparative study of the brain regions in which transgenic cells, nuclei, and granules were identified after a single injection of AAV also indicates a diffuse mode of AAV distribution. A summary plot (Figure 9) shows the comparative distribution in all brain regions where transgenes were detected at 1 week after the AAV injection, indicating the regional specificity of the AAV distribution.

As a result of a single injection of the recombinant AAVs of mammalian hypocampus, the maximum number of GFP+ cells was localized in the PTA and the anterior hypothalamic ventricle in the diencephalon (Figure 9). In the mesencephalon, most transgenes were found in the cells of DMT and a somewhat smaller number was in DLT (Figure 9). The maximum colocalization with the early neuronal differentiation protein HuCD was recorded from the same areas (Figure 9). The presence of the maximum number of GFP+ and HuCD+/GFP+ cells in PTA and the anterior hypothalamic ventricle is associated with a high density of distribution of cells in these diencephalon regions and suggests the efficient distribution of AAV vectors using the anterograde intracellular transport. The presence of a large number of GFP+ cells in DMT and DLT, including those co-expressing HuCD, indicates the distribution of AAV as a result of post-traumatic neurogenic events observed in the post-injection area. A large number of GFP+ cells with a relatively low co-expression of GFP+/HuCD+ is characteristic of the postcommissural area. A large number of GFP+ elements, which are granules of subcellular size, and a relatively small number of neurons were detected in this zone (Figure 9). The minimum number of GFP+ and GFP+/HuCD+ cells was found in MRF (Figure 9). This area of the brainstem is characterized by the lowest density of distribution of cells, which determines the total low number of both GFP+ cells and young neuroblasts. We also suggest that the distribution of the AAV vector using the retrograde intracellular transport is somewhat limited in comparison with the anterograde direction, which determines the low content of GFP+ cells in this area.

### 3.5. The Use of AdV and AAV in a Fish Model for Gene Therapy

The presented results may be important for gene therapy aimed at treating or preventing various human diseases. Among the best-known approaches, of particular note are the introduction of a new gene to restore the wild-type phenotype (gene supplements), correction of altered genome (gene editing), and inhibition/inactivation of the mutated gene responsible for the harmful phenotype. To date, several mechanisms have been proposed for introducing the desired genetic material into the target cell, such as physical methods (electroporation, microinjection, etc.), chemical methods (non-viral vectors), and viral vectors [44,45]. Viral vectors are of great clinical interest due to their high efficiency in the delivery of genetic material to the nucleus of the host cell. A simple and effective means for delivering genetic material to cells and living organisms is a valuable asset for research aimed at the regulated expression of certain gene products, vaccine delivery, and gene targeting. At present, the cellular transduction into teleost fish cells consists, as a rule, in ion permeability and electroporation solutions in cell culture and the use of direct contact with animal plasmid DNA [1,60]. The results of studies have shown the potential advantages of using viral vectors for transfer into fish cells both in vitro and in vivo with the expression of the transgenes. Nevertheless, the efficiency of in vivo transduction was found to be much lower than that recorded for the same carriers in laboratory strains of rats and mice [44,56].

In studies on vertebrates, adenoviral vectors have been found to cause less toxic effects than some other viruses, but they are not inert with respect to innate [78] or adaptive immunity, and they may also affect intracellular activity [79]. According to some reports, neurotoxic effects are observed in the case of systemic delivery of adenovirus or through direct injection into the central nervous system, as well as into the subretinal space of the mice retina [80,81].

The viral gene transfer in zebrafish has been achieved using baculoviruses, rabies virus, and Sindbis virus [41,82]. However, these vectors have practical disadvantages such as toxicity (in case of Sindbis), complicated procedures for the production and modification of viruses (rabies and baculoviruses), and the difficulty of obtaining high titers (rabies). The use of pseudotyped lettiviruses or murine leukemia virus is suggested as a potential way to circumvent these problems [83]. Another class of viral vectors with beneficial properties is modified herpes simplex viruses 1 (HSV-1) [84]. Although HSV-1 can infect zebrafish [85], vectors derived from HSV-1 have not yet been investigated as tools for introducing transgenes into fish neurons.

It is believed that a possible alternative approach to rapid gene transfer is electroporation based on short electrical pulses to temporarily permeate the plasma membrane and transfer nucleic acids into cells [86]. This method does not require the production of specialized vectors, is cost-effective, and has additional advantages [87]. Electroporation is a popular technique for manipulating neurons during development [88] and has been used in studies on different species [86,87], including zebrafish [89,90]. However, despite promising reports [87,91], electroporation is not a commonly applied method for injecting transgenes directly into spatially restricted populations of neurons in the adult brain.

The presented results show that the recombinant AAV vector after a single injection into the mesencephalon of juvenile chum salmon, *O. keta*, did not have a noticeable toxic effect on the fish brain cells in the study period, which makes it possible to use it as an effective tool for delivering genetic material to the brain cells of juvenile chum salmon as a potential model for neurogenic research. In contrast to experiments on zebrafish, the results of injecting AAV into the brain of juvenile salmon have shown efficient incorporation into both neurons and non-neuronal cells. The high proliferative neurogenic potential of the brain in salmonids and their capability of *fetalization* (delay of the embryonic structure in the development of the brain) are possibly the key issues in this process. Thus, based on the initial step of investigations that require subsequent experimental confirmation, we are planning to further study the neuronal and non-neuronal phenotypes of AAV-expressing cells in the brain of juvenile salmonids.

## 4. Material and Methods

### 4.1. Experimental Animals

We used 50 one-year-old juveniles of the Pacific chum salmon, *Oncorhynchus keta*, with a body length of 13–15.5 cm and a weight of 45–55 g. The animals were obtained from the Ryazanovka experimental fish hatchery in 2020. The animals were kept in a tank with aerated fresh water at a temperature of 15–16 °C and fed once a day. The light/dark cycle was set at 14/10 h. The content of dissolved oxygen in water was 7–10 mg/dm^3^, which corresponds to normal saturation. All experimental manipulations with animals were carried out in accordance with the rules of the charter of the Zhirmunsky National Scientific Center of Marine Biology (NSCMB) FEB RAS, and the Commission on the biomedical ethics, the Resource Center of the NSCMB FEB RAS, which regulates the humane treatment of experimental animals (approval no. 1-16042l from Meeting No. 1 of the Commission on the biomedical ethics of NSCMB FEB RAS, 16 April 2021).

### 4.2. Injection of Recombinant Adeno-Associated Viruses

We used ready-to-image recombinant adeno-associated viruses of the mouse hippocampus AAV1.Camc2a.GCaMP6f.WPRE.bGHpA (Inscopix, Palo Alto, CA, USA). Packaging, cleaning, and determination of vector titers were performed by Stanford University (Inscopix, USA). Recombinant vectors were purified using the CsCl precipitation method, and genomic copy titers were determined as previously described [92]. Injection titers were optimized for concentration and amounted to 1.68E13 µg/mL, which was functionally confirmed for calcium imaging of pyramidal neurons in the dorsal CA1 mouse hippocampus.

Animals were anesthetized in a cuvette containing 0.01% ethyl-3-aminobenzoate methanesulfonate (MS222) (Sigma, St. Louis, MO, USA, Cat. No. WXBC9102V) for 5 min at room temperature. After anesthesia, 0.2 µL of recombinant AAV solution in PBS (*n* = 5 for each group) was injected into the right part of the tegmentum using a Hamilton syringe (*n* = 5 for each group) according to the previously described technique [22]. Control animals received 0.2 μL 0.1 M PBS (*n* = 5). Immediately after injury, the animals were released into the aquarium for recovery and further monitoring.

### 4.3. Sample Preparation

After the intracranial injection into the mesencephalic tegmentum, the video monitoring of changes in motor and behavioral activity in fish in the experimental group was conducted for 1 h. After 1 week, the animals were removed from the experiment and euthanized by rapid decapitation. The brain was prefixed in a 4% paraphormaldehyde solution (PFA, BioChemica, Cambridge, MA, USA; Cat. No. A3813.1000; lot 31000997) prepared in 0.1 M phosphate buffer (Tocris Bioscience, Minneapolis, MN, USA; Cat. No. 5564, Batch No.: 5) (pH 7.2). After prefixation, the brain was removed from the cranial cavity and fixed in the same solution for 2 h at a temperature of 4 °C. Then, it was washed in a 30% sucrose solution at 4 °C for two days, with a five-fold change of the solution. Serial frontal 50-μm sections of fish brain were cut on a freezing microtome (Cryo-star HM 560 MV, Thermo Scientific, Walldorf, USA) and mounted on polylysine slides (Biovitrum, St. Petersburg, Russia).

### 4.4. Immunofluorescence Labeling

To identify the neuron-specific HuCD protein on brain slices containing AAV-labeled cells, we performed labeling with the corresponding primary mouse antibodies from Chemicon (clone: AD2.38; Chemicon Billerica, MA, USA) at a dilution of 1:200 in accordance with previously published data [22]. Sections were preincubated in PBS supplemented with 10% non-immune horse serum, 0.01% Tween 20 (Sigma, St. Louis, MO, USA), and 0.1% BSA (Sigma, St. Louis, MO, USA) for 30 min at room temperature. Then, the sections were incubated with primary antibodies at 4 °C for 48 h. After a short wash in PBS, the sections were incubated with a donkey anti-mouse Ig secondary antibody conjugated to Alexa 546 (Invitrogen, New York, SA, dilution 1: 200). To calculate the percentage of immunopositive neurons, in addition to the label to HuCD, the nuclei of all cells in the section were stained with DAPI solution (Invitrogen, New York, USA, D9542, final dilution of DAPI 0.01 μg/mL in PBS). Negative control was in the absence of secondary antibodies; the preparations were embedded in glycerol and contoured with varnish.

### 4.5. Microscopy

For visualization and morphological analysis, a motorized inverted microscope of research grade with a fluorescent module and an attachment for improved contrasting was used for visualization and morphological analysis when working with Axiovert 200 M luminescence with an ApoTome module (Carl Zeiss, Oberkochen, Germany). For a more detailed study of the tegmental area in three-dimensional space using a multispectral (with several fluorochromes) mode of operation, GFP/HuCD/DAPI colocalization was studied using a LSM 780 NLO confocal laser scanning microscope with a high-resolution Airiskan module (Carl Zeiss, Jena, Germany).

To study the micrographs of the preparations, the analysis of the material was carried out using the Axio Vision software (Carl Zeiss, Germany). Measurements were performed with several objective lenses at magnifications of 10×, 20×, and 40× in 10 randomly selected fields of view for each study area. The number of immunolabeled cells in the field of view was counted at a magnification of 200×. Morphometric analysis of the parameters of cell bodies (measurement of the greater and lesser diameters of neuron soma) was carried out using the Axio Vision microscope software. Counting was performed in 10 randomly selected measured areas (1 microscopic field of view was 0.12 mm^2^) at a 200× magnification for each animal. The average value was determined by averaging the proportions obtained from five animals.

### 4.6. Statistical Analysis

Prior to the experiments, we performed a statistical analysis based on the variations in the measured parameters in our previous research [22] and established that we needed a group of at least 4 animals to achieve the statistical confidence at 95%. To make sure that we reach a group size of 4 and, at the same time, reduce the use of animals to a minimum, we aimed for a total of 5 animals per experimental group.

Quantitative assessment of cells was carried out on a separately selected view field at a magnification of 10× of the objective lens and 20× of the eyepiece. All data analysis was performed using a blind test to reduce experimenter’s bias. To rank the labeled elements according to size groups, all data on measured cells/nuclei were divided into non-overlapping size groups (Table 1) and presented as Mean ± Standard Deviation of the mean (M ± SD). One-way analysis of variance (ANOVA) was used to assess intergroup differences when comparing the number of GCaMP6m-GFP-labeled, HuCD-immunopositive cells, and colocalization of these markers. Values at *p* < 0.05 and *p* < 0.01 were considered statistically significant. All quantitative results in the present study were analyzed in the SPSS software (version 16.0; SPSS Inc., Chicago, IL, USA). Quantitative processing of morphometric data of IHC labeling was performed using the Statistica 12, Microsoft Excel 2010, and STATA (StataCorp. 2012, Stata Statistical Software: Release 12) software packages.

## Figures and Tables

**Figure 1 ijms-22-05661-f001:**
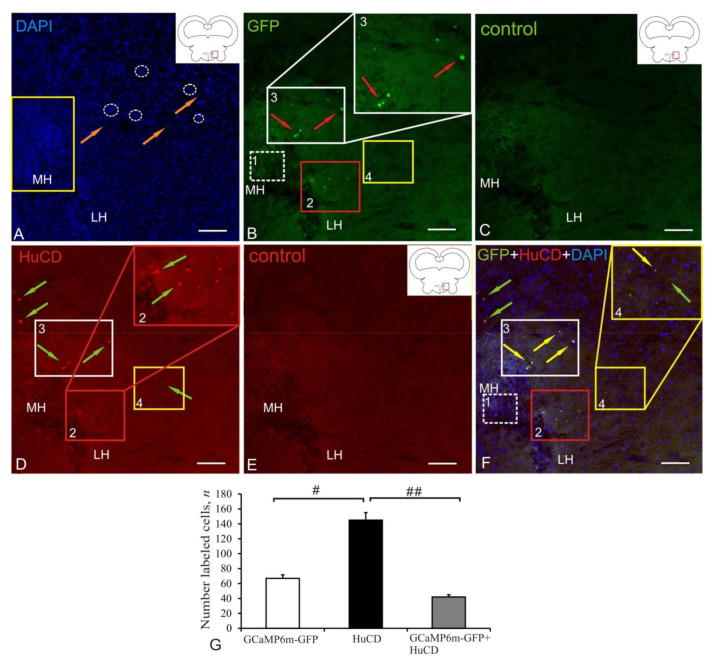
Z-stacks representing HuCD immunolabeling in the anterior hypothalamic ventricle of juvenile chum salmon, *O. keta*, at 1 week after a single injection of recombinant AAV in the mesencephalic tegmentum area. (**A**) DAPI staining; the pictogram shows the area of the anterior hypothalamus (outlined by red rectangle); the yellow rectangle outlines an aggregation of nuclei of the medial hypothalamus (MH), stained nuclei (orange arrows); LH is the lateral hypothalamus; constitutive clusters of nuclei are outlined by the white oval dotted line. (**B**) Expression of the green fluorescent protein (GFP) in hypothalamic cells (red arrows); the white dashed rectangle outlines cluster 1 in the medial zone; other boxes show clusters of 2, 4, and 3 (with inset at higher magnification) GFP+ cells in the lateral zone of the hypothalamus. (**C**) Control brain sections; in the animals that received an injection of 0.1% PBS, GCaMP6m-GFP was not labeled. (**D**) Immunofluorescence of the HuCD protein in hypothalamic neurons (green arrows); cluster 2 is shown with inset at higher magnification. (**E**) Control brain sections; in the animals with the lack of secondary antibodies, no HuCD immunolabeling was detected. (**F**) Superposition of three channels of DAPI/GFP/HuCD staining, showing the areas of GFP/HuCD colocalization in neurons (in the red rectangle). Laser scanning confocal microscopy. Scale bar: 200 µm. (**G**) Results of one-way analysis of variance (ANOVA test) showing the comparative distribution of labeled cells and nuclei (M ± SD, where M is the mean and SD is the standard deviation) of the anterior hypothalamic ventricle of chum salmon. Significant intergroup differences were found between groups of GCaMP6m-GFP+ and HuCD+ cells (# < 0.05), as well as between GCaMP6m-GFP+ and GCaMP6m-GFP+/HuCD+ cells (## < 0.01) (*n* = 5 in each group).

**Figure 2 ijms-22-05661-f002:**
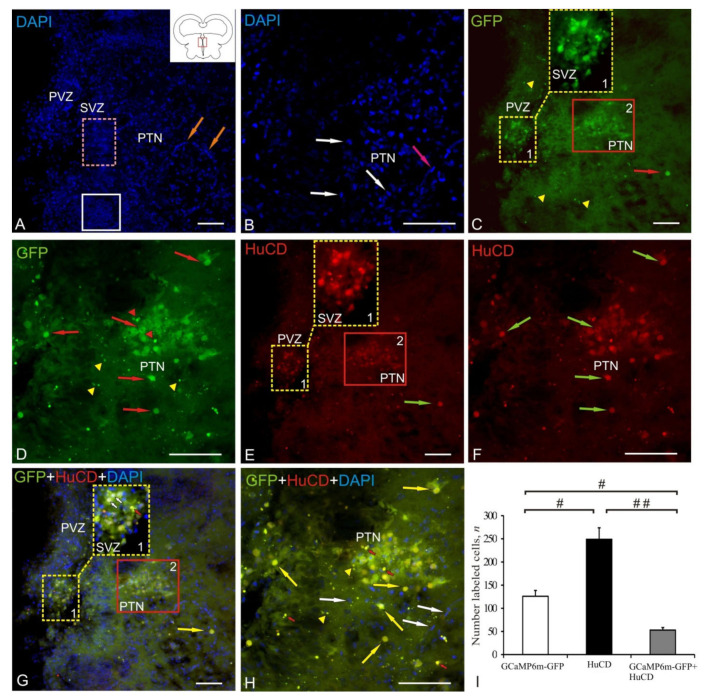
Z-stacks representing HuCD immunolabeling in the posterior tuberculum area of juvenile chum salmon, *O. keta*, at 1 week after a single injection of recombinant AAV into the mesencephalic tegmentum area. (**A**) DAPI staining, the pictogram shows the posterior tuberal area, morphologically heterogeneous nuclei forming dense clusters (in a white square), as well as multilayer clusters (outlined by pink dotted rectangle), abundantly vascularized (orange arrows), PVZ—periventricular zone, SVZ—subventricular zone, PTN—posterior tuberal nucleus. (**B**) PTN at higher magnification, heterogeneous nuclei with numerous nucleoli (white arrows), elongated nuclei of migrating cells (pink arrow). (**C**) Expression of green fluorescent protein GFP in PTA cells (red arrows), aggregation of GFP+ cells in SVZ (population № 1, yellow dashed rectangle) of heterogeneous composition (inset at high magnification), GFP+ neurons and nuclei in PTN (population № 2 in red rectangle), GFP+ granules (yellow arrowheads). (**D**) PTN at higher magnification, GFP+ nuclei (red triangular arrows). (**E**) Immunofluorescence of HuCD protein in PTA neurons (green arrows), population № 1 is shown with inset at higher magnification. (**F**) PTN at higher magnification. (**G**) Superposition of three DAPI/GFP/HuCD staining channels in PTA (populations № 1 and 2), demonstrating colocalization of GFP/HuCD in neurons (white arrows in the inset, population № 1) and nuclei (red arrow in the inset, population № 1) or PTN neurons (yellow arrow). (**H**) PTN at higher magnification, DAPI stained nuclei (white arrows), GFP+ granules (yellow arrowheads). Laser scanning confocal microscopy. Scale bars: (**A**,**C**,**E**,**G**)—200 µm; (**B**,**D**,**F**,**H**)—50 µms. (**I**) ANOVA analysis results showing the comparative distribution of labeled cells and nuclei (M ± SD, where M is the mean and SD is the standard deviation) of the posterior tuberal area of the chum salmon. Significant intergroup differences were found between the groups of GCaMP6m-GFP+ and HuCD+, between GCaMP6m-GFP+ and GCaMP6m-GFP+/HuCD+ (# < 0.05) cells; and between the groups of HuCD+ and GCaMP6m-GFP+/HuCD+ (## < 0.01) cells (*n* = 5 in each group).

**Figure 3 ijms-22-05661-f003:**
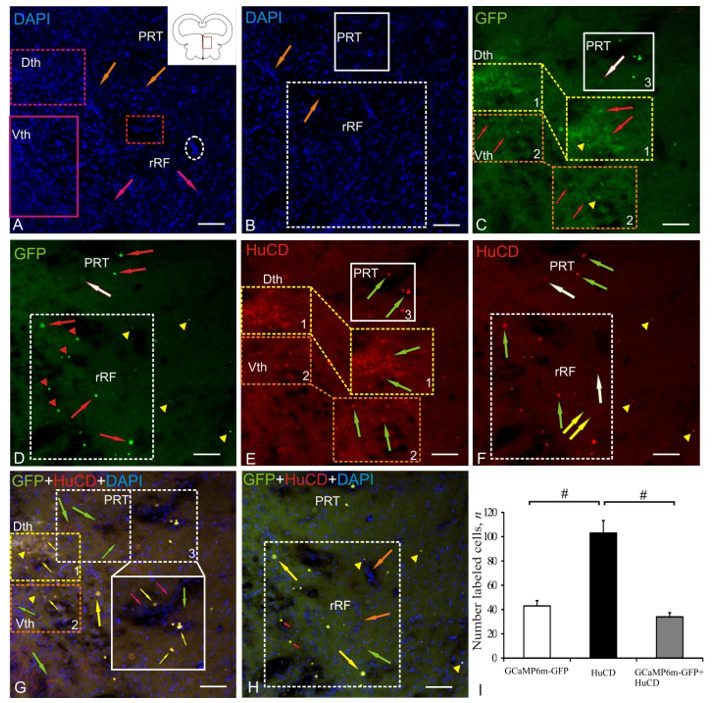
Z-stacks representing HuCD immunolabeling in the thalamus of juvenile chum salmon, *O. keta*, at 1 week after a single injection of recombinant AAV into the mesencephalic tegmentum area. (**A**) DAPI staining, the pictogram shows the thalamus area (in a red rectangle), rectangles outline clusters of small rounded nuclei, Dth—dorsal thalamus, Vth—ventral thalamus, DAPI-stained microvessel fragments (orange arrows), elongated nuclei (pink arrows), forming heterogeneous complexes with oval cells (outlined by red rectangle), as well as aggregations of nuclei (outlined by white dotted oval), forming parenchymal neurogenic niches. (**B**) DAPI-stained clusters of nuclei (outlined by white square) forming a complex of pretectal nuclei (PRT), ventral in the dotted square are thalamic zones of the reticular formation (rRF). (**C**) Expression of green fluorescent protein GFP in cells of the periventricular thalamus, an aggregation of GFP+ cells (red arrows) and granules (yellow arrowheads) in Dth (population № 1 outlined by yellow dashed rectangle), in Vth (population № 2 outlined by orange dotted rectangle), in PRT (population № 3 in the white rectangle), a weakly GFP-labeled cell (white arrow). (**D**) GFP expression in PRT cells (green arrows), rRF nuclei (red arrowheads), and granules (yellow arrowheads). (**E**) Immunofluorescence of the HuCD protein in neurons of the periventricular thalamus (green arrows), other designations are as in (**C**). (**F**) HuCD in PRT neurons (green arrows—intensely labeled, white—weakly labeled), in rRF (yellow arrows indicate weakly labeled differentiated neurons, yellow arrowheads indicate small neurons). (**G**) Superposition of three channels of DAPI/GFP/HuCD staining in cells (yellow arrows); Dth and Vth (populations № 1, 2), granules (yellow arrowheads), GFP–/HuCD+ neurons (red arrows), in PRT (population № 3) HuCD+/DAPI+/GFP– large neurons (green arrows) adjoined the population of GFP+/HuCD+ cells and nuclei on one side (in a white dotted rectangle), and on the other, to the vascular sinus (inset, outlined by white rectangle) with migration patterns (pink arrows). (**H**) In rRF, DAPI/GFP/HuCD colocalization patterns (outlined by white dotted square) in GFP+/HuCD+ cells (yellow arrows), nuclei (red arrows), and granules (yellow arrowheads). (**I**) Results of ANOVA analysis showing the comparative distribution of labeled cells and nuclei (M ± SD, where M is the mean and SD is the standard deviation) in the thalamus of juvenile chum salmon, *O. keta*. Significant intergroup differences (# < 0.05) were found between the groups of GCaMP6m-GFP+ and GCaMP6m-GFP+/HuCD+ cells and the groups of HuCD+ and GCaMP6m-GFP+/HuCD+ cells (*n* = 5 in each group).

**Figure 4 ijms-22-05661-f004:**
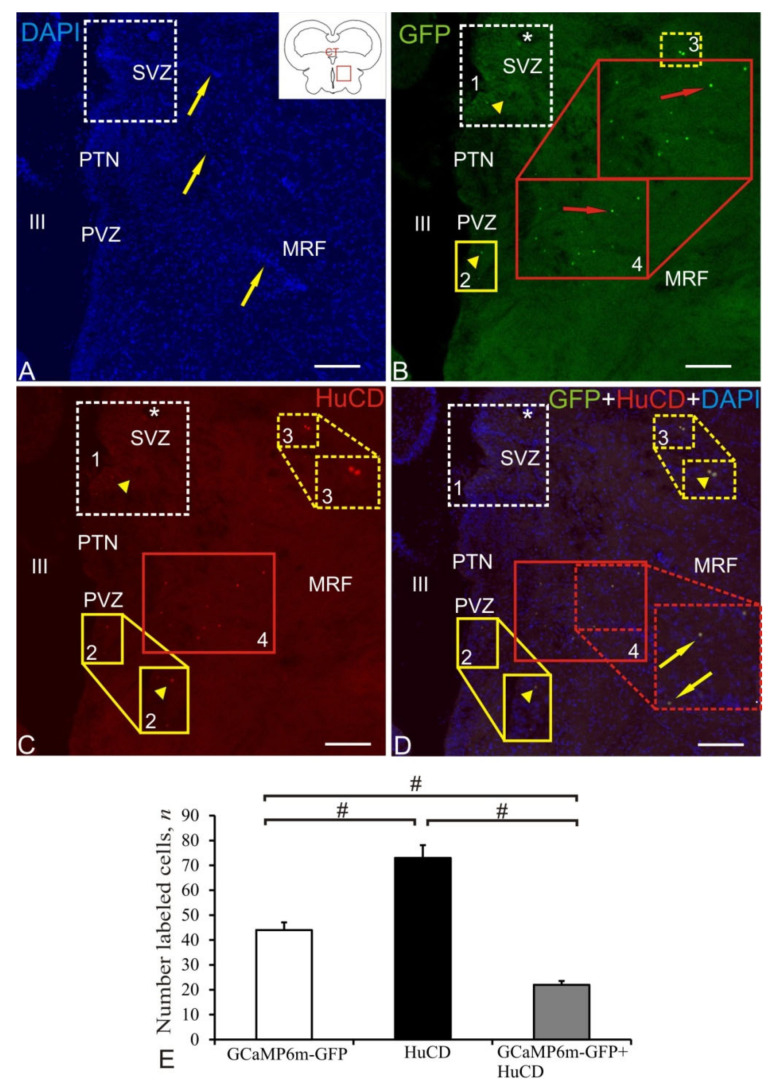
Z-stacks representing HuCD immunolabeling in the postcommissural region of the basal mesencephalon of juvenile chum salmon, *O. keta*, at 1 week after a single injection of recombinant AAV into the mesencephalic tegmentum region. (**A**) DAPI staining, the pictogram shows the postcommissural region of the basal mesencephalon (outlined by the red rectangle), the dashed rectangle outlines the dorsal clusters, the nuclei are indicated by yellow arrows, PVZ—the periventricular zone, SVZ—the subventricular zone, PTN—the posterior tuberal nucleus, the MRF—mesencephalic reticular formation, III—hypothalamic ventricle. (**B**) Expression of GFP in cells (red arrows), dorso-medial posterior tuberal group (population № 1) (outlined by white dotted line), GFP+ granules (yellow arrowheads), perventricular aggregation (population № 2), an asterisk shows a GFP- nucleus aggregation, individual GFP+ cells in MRF, (population № 3), an aggregation of intensely fluorescent cells (population № 4) is outlined by red square, the inset shows a higher magnification. (**C**) Immunofluorescence of HuCD, designation as in B, neurons of populations № 2 and 3 are shown at higher magnification (inset). (**D**) Superposition of three DAPI/GFP/HuCD staining channels, showing areas of GFP/HuCD colocalization in neurons (yellow arrows), neurons of populations № 2, 3, and 4 are shown at higher magnification (inset). Laser scanning confocal microscopy. Scale bar: 200 µm. (**E**) Results of ANOVA analysis showing the comparative distribution of labeled cells and nuclei (M ± SD, where M is the mean and SD is the standard deviation) in the postcommissural area of the chum salmon. Significant intergroup differences (# < 0.05) were found between the groups of GCaMP6m-GFP+ and HuCD+ cells, GCaMP6m-GFP+ and GCaMP6m-GFP+/HuCD+ cells, and between the groups of HuCD+ and GCaMP6m-GFP+/HuCD+ cells (*n* = 5 in each group).

**Figure 5 ijms-22-05661-f005:**
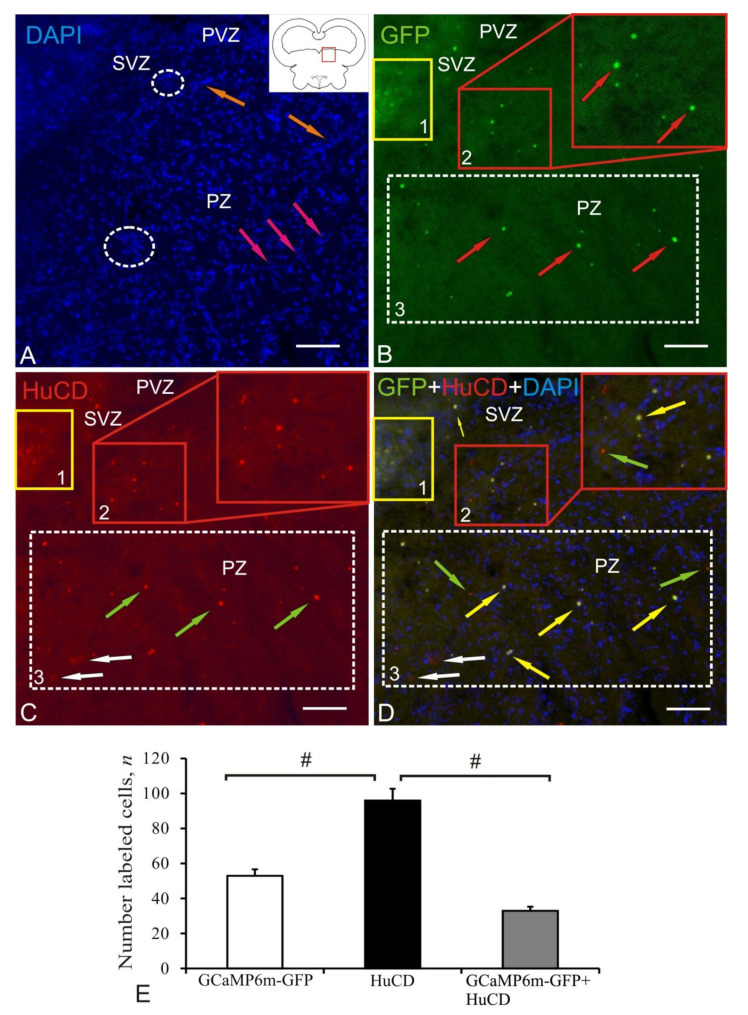
Z-stacks representing HuCD immunolabeling in the dorso-medial tegmentum of juvenile chum salmon, *O. keta*, 1 week after a single injection of recombinant AAV into the mesencephalic tegmentum area. (**A**) DAPI staining, the pictogram shows the area of the dorso-medial tegmentum (outlined by red rectangle, resting nuclei are indicated by orange arrows, migrating ones by pink arrows), clusters of nuclei are outlined by the white dashed oval, PZ is the parenchymal zone, and the other designations are as in Figure 4A. (**B**) GFP expression in cells (red arrows) of population № 1 (outlined by the yellow rectangle), population № 2 (outlined by the red rectangle, greater magnification in the inset), and population № 3 (outlined by the white dashed rectangle). (**C**) Immunofluorescence of HuCD (populations № 1–3) in small intensely labeled cells (green arrows) and large weakly labeled cells (white arrows). (**D**) Superposition of three DAPI/GFP/HuCD staining channels, showing areas of GFP/HuCD colocalization in neurons of populations № 2 and 3 (yellow arrows), cells without marker colocalization (green arrows), in populations № 3 large weakly HuCD-labeled cells (white arrows). Laser scanning confocal microscopy. Scale bar: 200 µm. (**E**) Results of ANOVA analysis showing the comparative distribution of labeled cells and nuclei (M ± SD, where M is the mean and SD is the standard deviation) of the dorso-medial tegmentum of chum salmon. Significant intergroup differences (# < 0.05) were found between the groups of GCaMP6m-GFP+ and HuCD+ cells and the groups of HuCD+ and GCaMP6m-GFP+/HuCD+ cells (*n* = 5 in each group).

**Figure 6 ijms-22-05661-f006:**
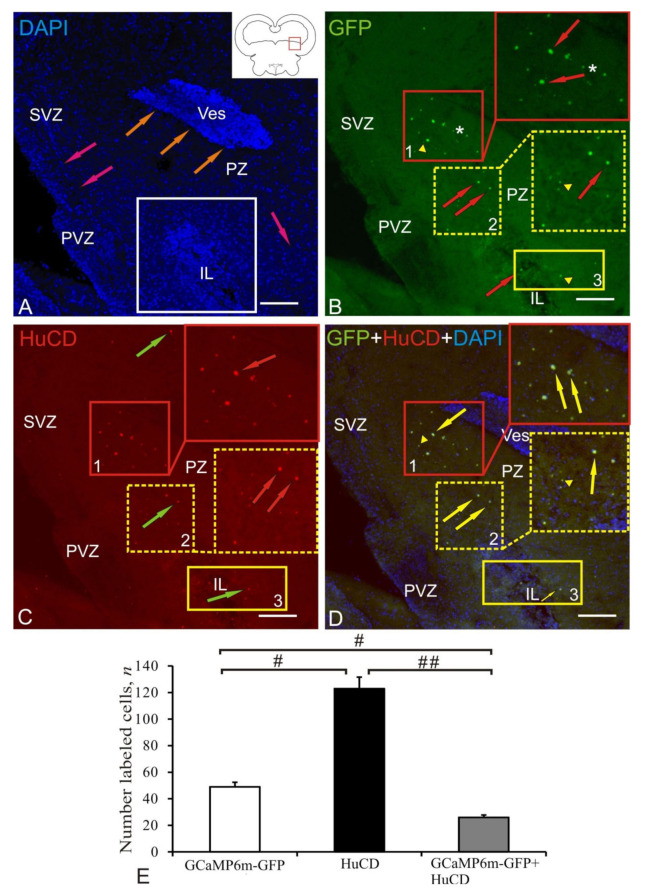
Z-stacks representing HuCD immunolabeling in the dorso-lateral tegmentum of juvenile chum salmon, *O. keta*, at 1 week after a single injection of recombinant AAV into the mesencephalic tegmentum area. (**A**) DAPI staining, the pictogram shows the area of the dorso-lateral tegmentum (outlined by red rectangle), pink arrows indicate nuclei, orange arrows indicate the dilated blood sinus (Ves), the post-injection cavity (IL) is outlined by the white square. (**B**) Expression of GFP in cell populations: outlined by the red rectangle (population № 1, an enlarged fragment in the inset), single GFP+ cells (red arrows), GFP+ nuclei (white asterisk), GFP+ granules (yellow arrowhead), population № 2, on inset enlarged fragment (in the yellow dotted rectangle), population № 3 (in the yellow rectangle). (**C**) Immunofluorescence of HuCD in populations № 1–3 of neurons (green arrows, red arrows in the inset, other designations as in (**B**)). (**D**) Superposition of three DAPI/GFP HuCD staining channels, showing areas of GFP/HuCD colocalization in neurons (yellow arrows). Laser scanning confocal microscopy. Scale bar: 200 µm. (**E**) Results of ANOVA analysis showing the comparative distribution of labeled cells and nuclei (M ± SD, where M is the mean and SD is the standard deviation) of the dorso-medial tegmentum of chum salmon. Significant intergroup differences were found between the groups of GCaMP6m-GFP+ and HuCD+ cells, the GCaMP6m-GFP+ and GCaMP6m-GFP+/HuCD+ cells (# < 0.05), and between the groups of HuCD+ and GCaMP6m-GFP+/HuCD+ (## < 0.01) cells (*n* = 5 in each group).

**Figure 7 ijms-22-05661-f007:**
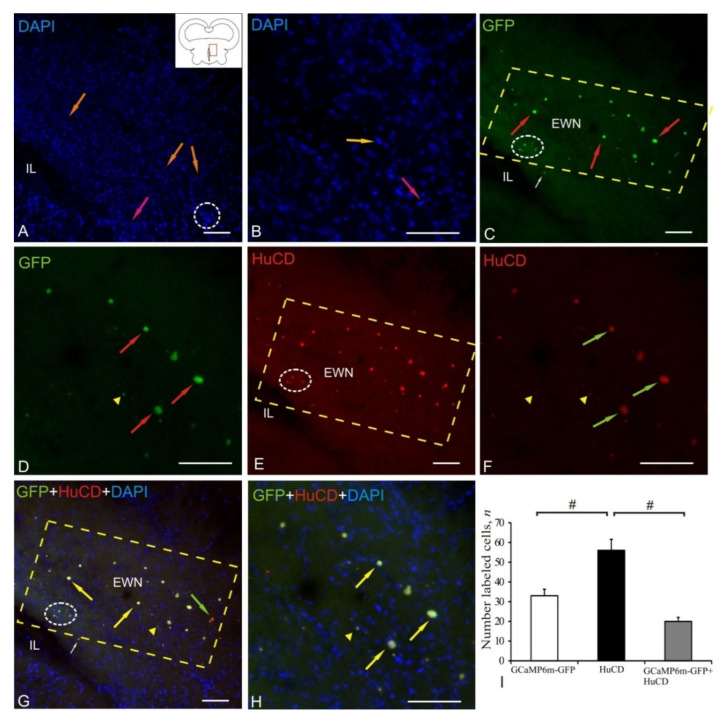
Z-stacks representing HuCD immunolabeling in the Edinger–Westphal nucleus of juvenile chum salmon, *O. keta*, at 1 week after a single injection of recombinant AAV into the mesencephalic tegmentum area. (**A**) DAPI staining, the pictogram shows the area of the Edinger–Westphal nucleus (outlined by red rectangle) adjacent to the post-injection cavity (IL), single aggregations of nuclei/cells forming neurogenic niches (outlined by white oval) and numerous oval nuclei (orange arrows), or elongated shape (pink arrows). (**B**) At higher magnification, (**C**) GFP expression in cells (red arrows), aggregation of GFP+ granules (in the white oval), single GFP+ nuclei (white arrow). (**D**) At higher magnification, GFP+ granules (yellow arrowhead). (**E**) Immunofluorescence of the HuCD protein in EWN neurons, designations as in B. (**F**) EWN at higher magnification, HuCD+ neuron (red arrows). (**G**) Superposition of three DAPI/GFP/HuCD staining channels showing the areas of GFP/HuCD colocalization in neurons (yellow arrows), nucleus (white arrow), and GFP–/HuCD+ neurons (green arrow). (**H**) Colocalization in EWN at higher magnification. (**I**) Results of ANOVA analysis showing the comparative distribution of labeled cells and nuclei (M ± SD, where M is the mean and SD is the standard deviation) in the Edinger–Westphal nucleus of juvenile chum salmon. Significant intergroup differences (# < 0.05) were found between the groups of GCaMP6m-GFP+ and GCaMP6m-GFP+/HuCD+ cells and between groups of the HuCD+ and GCaMP6m-GFP+/HuCD+ cells (*n* = 5 in each group).

**Figure 8 ijms-22-05661-f008:**
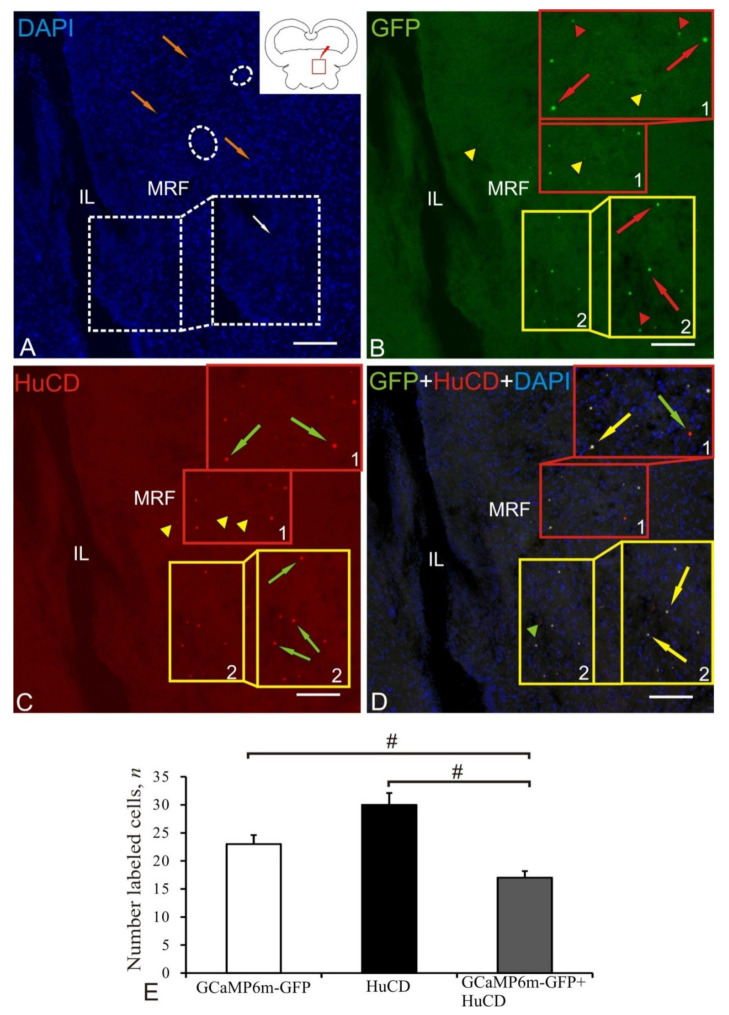
Z-stacks representing HuCD immunolabeling in the mesencephalic reticular formation of juvenile chum salmon, *O. keta*, at 1 week after a single injection of recombinant AAV into the mesencephalic tegmentum. (**A**) DAPI staining, a diagram of the AAV injection area (indicated by red zigzag) and the distribution of GFP expressing cells in the MRF is shown in the pictogram (outlined by the red square), labeled cell nuclei (orange arrows) and their clusters (outlined by the white oval), along the injection lumen (IL), invaginations of the parenchymal brain tissue were revealed, with heterogeneous clusters of stained nuclei (outlined by the white dotted rectangle), forming reactive neurogenic zones (dotted inset). (**B**) GFP expression in population № 1 (outlined by the red rectangle) containing GFP+ neurons (shown in the inset, red arrows), GFP+ nuclei (red arrowheads) and GFP+ granules (yellow arrowheads) and population № 2 (yellow rectangle, inset). (**C**) HuCD immunofluorescence in populations № 1 and 2, neurons are indicated by green arrows in the inset, yellow arrowheads indicate granules. (**D**) Superposition of three channels of DAPI/GFP/HuCD staining in population № 1 of neurons (yellow arrow) and GFP-/HuCD+ cells (green arrow), in population № 2 (inset in yellow rectangle), GFP–/HuCD+ granules (green arrowheads). (**E**) Results of ANOVA analysis showing the comparative distribution of labeled cells and nuclei (M ± SD, where M is the mean and SD is the standard deviation) in the MRF of juvenile chum salmon. Significant intergroup differences (# < 0.05) were found between the groups of GCaMP6m-GFP+ and GCaMP6m-GFP+/HuCD+ cells and the groups of HuCD+ and GCaMP6m-GFP+/HuCD+ cells (*n* = 5 in each group).

**Figure 9 ijms-22-05661-f009:**
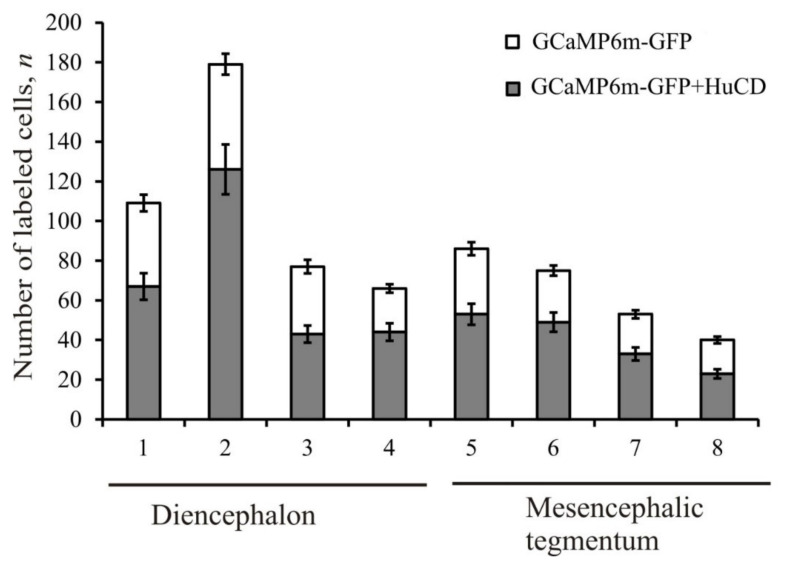
Comparative distribution (M ± SD) of GCaMP6m-GFP expressing cells and cells containing double GCaMP6m-GFP+/HuCD labeling in the mesencephalon and diencephalon of juvenile chum salmon, *O. keta*, at 1 week after a single injection of recombinant AAV mammalian hypocampus. Numerals along the X axis are as follows: (1) anterior hypothalamic ventricle; (2) posterior-tuberal area, 3—dorsal thalamus, 4—postcommissural area, 5—dorso-medial tegmentum, 6—dorso-lateral tegmentum, 7—Edinger–Westphal nucleus, 8—mesencephalic reticular formation.

**Table 1 ijms-22-05661-t001:** Morphometric parameters * of labeled neurons (M ± SD) in the mesencephalic tegmentum of the juvenile chum salmon, *O. keta*, one week after a single injection of the adenoviral vector.

Brain Area	DAPI Size of Nuclei/ Cells, μm	AAV Size of Cells, Nuclei, Granules, μm	Optical Density, UOD	HuCD Cell Size, μm	Optical Density, UOD	AAV + HuCD Size of Cells, Nuclei, Granules, μm	Optical Density, UOD
**Anterior hypothalamic ventricle**	4.9 ± 0.3/3.4 ± 0.4 4.2 ± 0.1/3.1 ± 0.3 3.8 ± 0.1/2.9 ± 0.3 3.1 ± 0.4/2.8 ± 0.2	6.2 ± 0.6/6.2 ± 0.5 4.9 ± 0.3/4.6 ± 0.4 4.5 ± 0.1/3.6 ± 1.1 3.6 ± 0.2/3.1 ± 0.3 2.6 ± 0.4/2.3 ± 0.4	+++ +++ +++/++ ++ +/++	8.7 ± 0.4/6.0 ± 0.8 7.4 ± 0.4/5.8 ± 1.9 6.1 ± 0.3/5.1 ± 0.6 4.9 ± 0.3/4.4 ± 0.1 3.9 ± 0.3/3.4 ± 0.4	++ ++ +++ +++ ++/+++	6.5 ± 0.2/5.6 ± 0.4 4.8 ± 0.5/4.6 ± 0.6 3.8 ± 0.2/3.2 ± 0.5 3.2 ± 0.2/2.7 ± 0.3	++ +++ ++/+++ ++
**Posterior tuberculum area**	8.5 ± 0.7/6.0 ± 1.8 6.2 ± 0.3/4.3 ± 0.7 4.8 ± 0.4/3.9 ± 0.5	8.4 ± 0.9/6.0 ± 0.8 5.8 ± 1.1/4.5 ± 0.8 3.4 ± 0.4/2.7 ± 0.7 2.4 ± 0.3/2.0 ± 0.4	+++ ++ ++ +/++	8.2 ± 0.8/6.9 ± 0.8 6.5 ± 0.4/5.2 ± 0.4 3.9 ± 0.2/2.9 ± 0.4	++ +++ ++	8.1 ± 1.0/5.7 ± 1.4 6.3 ± 0.3/4.7 ± 0.7 4.4 ± 0.7/3.5 ± 0.5	++/+++ ++ +++
**Dorsal thalamus**	4.3 ± 0.2/3.2 ± 0.6 3.6 ± 0.2/2.8 ± 0.4 3.2 ± 0.2/2.6 ± 0.2	6.3 ± 0.3/5.4 ± 1.0 5.1 ± 0.4/4.0 ± 0.7 3.8 ± 0.4/3.4 ± 0.4	++/+++ +++ ++	7.2 ± 0.5/5.8 ± 0.9 6.0 ± 0.3/4.8 ± 0.6 4.8 ± 0.4/3.8 ± 0.7 3.0 ± 0.4/3.0 ± 0.3	++ ++/+++ +++ ++	8.1 ± 0.7/6.7 ± 1.4 6.9 ± 0.2/5.4 ± 0.9 5.8 ± 0.3/4.9 ± 0.4 4.9 ± 0.4/4.5 ± 0.4	++/+++ ++ +++/++ +++
**Postcommissural area**	5.3 ± 0.2/4.0 ± 0.5 4.5 ± 0.2/3.8 ± 0.4 3.7 ± 0.3/3.4 ± 0.4	8.8 ± 0.4/6.6 ± 0.2 5.2 ± 0.3/3.3 ± 0.8 3.0 ± 0.3/2.3 ± 0.3 2.4 ± 0.2/2.1 ± 0.3	+/++ +++ ++ +/++	7.8 ± 2.1/5.9 ± 2.6 4.6 ± 0.2/3.7 ± 0.4 3.7 ± 0.3/2.8 ± 0.5	+/++ +++ +++	5.7 ± 0.4/3.5 ± 0.5 4.6 ± 0.3/3.4 ± 0.4 3.8 ± 0.1/3.2 ± 0.5 2.3 ± 0.4/2.2 ± 0.3	++ + ++ +
**Dorso-medial tegmentum**	7.8 ± 0.4/5.0 ± 0.6 5.9 ± 0.6/4.4 ± 0.5 4.5 ± 0.5/3.6 ± 0.2	10.8 ± 0.3/8.0 ± 0.7 8.7 ± 0.1/6.6 ± 0.5 7.2 ± 0.4/5.5 ± 0.6 5.9 ± 0.5/4.7 ± 0.7 4.5 ± 0.4/3.9 ± 0.6 2.4 ± 0.4/2.2 ± 0.2	++ +++ +++ + +/++ +++	26.1 ± 1.3/13.4 ± 1.5 11.0 ± 0.7/7.3 ± 2.0 8.3 ± 0.6/6.4 ± 1.2 6.5 ± 0.6/5.2 ± 0.9 5.0 ± 0.1/4.6 ± 0.3	+ ++/+++ +++ + +/++	8.3 ± 0.7/6.4 ± 0.8 6.3 ± 0.8/5.1 ± 0.6 4.5 ± 0.2/3.8 ± 0.6	++ +++ +/++
**Dorso-lateral tegmentum**	4.8 ± 0.3/3.2 ± 0.5 4.2 ± 0.2/3.1 ± 0.3 3.7 ± 0.1/2.8 ± 0.3 3.2 ± 0.2/2.7 ± 0.5	5.9 ± 0.6/4.7 ± 0.2 4.9 ± 0.1/3.6 ± 0.4 4.4 ± 0.1/3.6 ± 0.4 3.6 ± 0.2/2.8 ± 0.3 2.9 ± 0.4/2.6 ± 0.3	+++ ++/+++ ++ ++ +/++	5.6 ± 0.3/4.2 ± 0.5 4.7 ± 0.2/3.6 ± 0.4 4.3 ± 0.2/3.3 ± 0.4 3.5 ± 0.2/2.8 ± 0.3	+++ +++ ++/+++ ++	5.5 ± 0.6/4.3 ± 0.8 4.5 ± 0.3/3.8 ± 0.6 3.9 ± 0.1/3.5 ± 0.2 2.9 ± 0.4/2.6 ± 0.3	+++ ++/+++ ++ +
**Edinger–Westphal nucleus**	4.0 ± 0.3/2.8 ± 0.8 3.4 ± 0.2/2.9 ± 0.3 2.8 ± 0.2/2.5 ± 0.3	10.2 ± 1/7.6 ± 0.6 6.4 ± 0.5/5.3 ± 0.6 4.5 ± 0.3/3.7 ± 0.4 3.5 ± 0.5/2.6 ± 0.5	+++ +++ ++/+++ ++	10.6 ± 1.2/7.7 ± 0.6 7.6 ± 0.4/6.4 ± 1.1 5.8 ± 0.7/4.5 ± 0.8 4.1 ± 0.3/3.4 ± 0.5	+++ +++/++ ++ ++	7.4 ± 0.3/6.2 ± 0.6 5.3 ± 0.5/4.2 ± 0.4 3.8 ± 0.3/2.8 ± 0.8	+++ +++/++ ++
**Mesencephalic reticular formation**	5.4 ± 0.3/3.8 ± 0.3 4.2 ± 0.3/3.3 ± 0.4 3.4 ± 0.3/2.7 ± 0.1	5.8 ± 0.3/4.4 ± 0.7 4.8 ± 0.3/3.9 ± 0.8 2.3 ± 0.1/2.2 ± 0.3	+++ ++/+++ +/++	5.7 ± 0.4/4.3 ± 0.7 4.7 ± 0.2/3.7 ± 0.7 4.0 ± 0.3/3.5 ± 0.4	+++ +++ ++/++	5.7 ± 0.5/4.5 ± 0.2 4.4 ± 0.3/4.1 ± 0.4	+++ ++

* Values of the greater and lesser diameters of nuclei or cell bodies. Values are arranged in descending order. Sizes of the labeled nuclei ranged from 2.5 to 3.5 μm [22]; the larger units are represented by variously sized cells.

**Table 2 ijms-22-05661-t002:** Proportion of HuCD-immunopositive and rAAV-labeled neurons in the mesencephalic tegmentum of juvenile chum salmon, *O. keta*, at one week after a single injection of the vector.

Brain Area	AAV, %	HuCD, %	AAV + HuCD, %
Anterior hypothalamic ventricle	26.4	57	16.5
Posterior tuberculum area	29.4	58.2	12.4
Dorsal thalamus	23.9	57.2	18.9
Postcommissural area	31.6	52.5	15.8
Dorso-medial tegmentum	29.1	52.7	18.1
Dorso-lateral tegmentum	24.7	62.1	13.1
Edinger–Westphal nucleus	30.3	51.2	18.3
Mesencephalic reticular formation	32.8	42.8	24.3

## Data Availability

Not applicable.

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
