# Peer review of "A Confocal Microscopic Study of Gene Transfer into the Mesencephalic Tegmentum of Juvenile Chum Salmon, Oncorhynchus keta, Using Mouse Adeno-Associated Viral Vectors"

_ijms, 2021, doi:10.3390/ijms22115661_

Round 1
Reviewer 1 Report
In this study, the authors intracranially injected GFP-containing rAAV vector into the mesencephalic tegmentum region of juvenile chum salmon. They studied the transduction of rAAV in the mouse hippocampus into brain cells of juvenile chum salmon and subsequently determined of the phenotype of rAAV-labeled cells by confocal microscopy. The results of the analysis showed partial colocalization of cells expressing green fluorescent AAV with red fluorescent HuCD protein. This study may provide clues to use AAV as a tool for delivering genetic material to the brain cells of juvenile chum salmon. However, the current design, writing and presentation of this study are not well. A numerous statements are confusing.
Some main comments:
- Adenovirus and Adeno-associated virus (AAV) are different viruses. It seems that the authors did not realize it. They explored AAV transduction in salmon brain, but they introduce adenovirus in Introduction section and mixed two in the following sections. And AdV is the abbreviation of Adenovirus.
- It would be good for the authors to have more information about AAV from literature. “which indicates the ability of hippocampal mammal adenoviruses to integrate into neurons of the central nervous system of fish with subsequent expression of viral proteins”. AAV rarely integrate its DNA into human genome.
- They used rAAV packaged with a calcium indicator of the latest generation GCaMP6m. I think GFP or eGFP would be much better. Their aim is to study the transduction and distribution of AAV in salmon brain. But GCaMP6m probably is expressed in calcium-abundant cells, which will not reflect the real efficiency or distribution of AAV in fish brain.
- How to explain the strong green or red background in all confocal images? The controls, no AAV injection and/or no second antibody in HuCD Immuno-staining, should be provided.
- In the statistic figures, using GCaMP6m-GFP instead of AAV would be better.
- In figure 2, ACEG(top)-BDFH(bottom) would be more clear. Also apply to Figures 3 and 7.
Minors:
- 1.68E + 13 μg/mL should be 1.68E13 μg/mL
- Please give more details about HuCD protein (such as full name, function) when first cited in the manuscript.
- Labeling the line number would make the comments easier.
- Need English native speaker to check the writing.
- Results 3.2, AVV to AAV.
Author Response
We would like to thank the Reviewers for the thoughtful and in-depth comments in our manuscript. Your suggestions and remarks have helped us to reflect on our paper and improve it. We appreciate your commitment and effort. We have carefully considered every comment, promptly accepted all the suggestions, and made the alterations as recommended:
Main comments:
- Adenovirus and Adeno-associated virus (AAV) are different viruses. It seems that the authors did not realize it. They explored AAV transduction in salmon brain, but they introduce adenovirus in Introduction section and mixed two in the following sections. And AdV is the abbreviation of Adenovirus.
The authors are aware that adenovirus and adeno-associated virus are different viruses. In the section Introduction, which contained inaccuracies on this issue, appropriate corrections were made and data on adeno-associated viruses were added.
- It would be good for the authors to have more information about AAV from literature. “which indicates the ability of hippocampal mammal adenoviruses to integrate into neurons of the central nervous system of fish with subsequent expression of viral proteins”. AAV rarely integrate its DNA into human genome.
Added data on adeno-associated viruses to the Introduction section
- They used rAAV packaged with a calcium indicator of the latest generation GCaMP6m. I think GFP or eGFP would be much better. Their aim is to study the transduction and distribution of AAV in salmon brain. But GCaMP6m probably is expressed in calcium-abundant cells, which will not reflect the real efficiency or distribution of AAV in fish brain.
We agree with this comment; however, the aim of our work was to conduct a confocal microscopic histological study of the patterns of GFP expression in the brains of salmon specifically as part of the rAAV + calcium indicator of the latest generation GCaMP6m. These data are necessary for our further optogenetic studies in vivo on salmonids.
- How to explain the strong green or red background in all confocal images? The controls, no AAV injection and/or no second antibody in HuCD Immuno-staining, should be provided.
We believe that the background is caused by the settings of the confocal microscope. In studies on mammals, in some cases, a fairly pronounced background is also observed (Hudry et al., 2019). As required by the reviewer, controls no AAV injection and / or no second antibody on HuCD immunostaining are provided
- In the statistic figures, using GCaMP6m-GFP instead of AAV would be better.
Changes applied
- In figure 2, ACEG(top)-BDFH(bottom) would be more clear. Also apply to Figures 3 and 7.
Unfortunately, the reformatting of Figures 2, 3 and 7 is associated with large changes in the corresponding part of the Results. Since Fig. 2A shows a general view, and in Fig. 2B shows the details of the brain structure shown in 2A (similarly in Figs. 3 and 7) at a higher magnification and this order of illustrations is closely related to the description in the corresponding part of the Results, we consider it inappropriate to make these changes. We kindly ask you to take these explanations into account.
Minors:
- 1.68E + 13 μg/mL should be 1.68E13 μg/mL
Changes applied
- Please give more details about HuCD protein (such as full name, function) when first cited in the manuscript.
In the Discussion section we add more information on the HuCD protein.
- Labeling the line number would make the comments easier.
Recommendations taken into account
- Need English native speaker to check the writing.
Recommendations taken into account
- Results 3.2, AVV to AAV.
Changes applied

Reviewer 2 Report
The authors asked the first feasibility and gene delivery of mouse recombinant adeno-associated viral vectors (rAAV) with a calcium indicator GCaMP6m in the brain of juvenile chum salmon, Oncorhynchus keta. Although the adenoviruses are known to be capable of infecting all classes of vertebrates including some fish, this attempt may have a significant potential for future vertebrate neuroscience, because of the present wide usage of rAAV in molecular neuroscience.
This paper is well described and discussed for new findings, but the overall purpose is obscure due to inconsistent usage of various genes, viral serotypes, injection places, species selection, and experimental environments compared with previous studies. The authors say that the purpose of this study to test the feasibility of rAAV in fish due to its importance in neurosciences, however, there are already several attempts in fish including cerebellum, and this study used different species and different brain regions. Such an experiment is important for comparative neuroscience, but a more careful explanation would be required to increase readability to scientific readers.
As the best finding in this study, the authors show the region-specific differential viral expression patterns: "different degrees of colocalization with early neuronal differentiation HuCD were found in several brain regions", and they are discussing that "proximity to the injection zone does not specifically affect the number of transduced cells". However, in mammalian or primate AAV studies, the proximity to the injection zone would really affect the expression. Also, this differentiation expression would be really affected with various environmental factors, e.g. temperature, haplotype of rAAV, incubation time (1 week only), neurogenic and cell type ratio, virus titer, viral diffusion ration in the brain, species specific expression, and inconsistent human technical issues. I agree that the statistics would convince the author's conclusion, but we know too many factors unconvinced in this study.
Therefore, this paper would be required to clarify the reasons:
1) Why the authors selected salmon for testing rAAV for fish?
2) Why selected GCaMP6m and did not see the expression in vivo as a calcium indicator?
3) Clarifying how the authors determined the suitable temperature, incubation time, viral titer, kinds of tissues, kind of the viral serotype, despite the first attempt with rAAV to salmon? The selection of serotype, temperature, and incubation time would be particularly crucial.
Author Response
Dear Reviewer, we thank you for your attentive attitude to our work and valuable comments that help to improve the content of our manuscript. Further, we provide answers to the comments and indicate what specific corrections were made to the text of the article.
- This paper is well described and discussed for new findings, but the overall purpose is obscure due to inconsistent usage of various genes, viral serotypes, injection places, species selection, and experimental environments compared with previous studies. The authors say that the purpose of this study to test the feasibility of rAAV in fish due to its importance in neurosciences, however, there are already several attempts in fish including cerebellum, and this study used different species and different brain regions. Such an experiment is important for comparative neuroscience, but a more careful explanation would be required to increase readability to scientific readers.
To improve the readability of the article, we have added additional data on the use of ААV in the Introduction. The data on the cerebellum are preliminary from our own research (Stukaneva et al., 2020). In general, according to the recommendation, additional explanations have been made in the Introduction section that improve the perception of the work by the readers and detail the purpose of this work.
- Why the authors selected salmon for testing rAAV for fish?
In the present study, juvenile Pacific salmon were selected as a model object to study the ability of AAV to transduce nerve tissue. Previous studies have shown the possibility of incorporating adenoviruses into other fish tissues and into the cell culture of salmonids. However, the data of Zhu et al. (2009) showed that adeno-associated viruses and lentiviruses reproduce poorly in the brain tissue of zebrafish. Based on our previous studies, which investigated the high neurogenic potential of the brain of juvenile salmonids (Pushchina et al., 2020; 2021), we hypothesized that a high production of neurons in the brain of juvenile salmonids may provide better rAAV transduction in their brains.
- Why selected GCaMP6m and did not see the expression in vivo as a calcium indicator?
This work is a neurohistological study of the ability of GCaMP6m containing GFP to integrate into brain cells of juvenile chum salmon and subsequent determination of the phenotypes of GCaMP6m-GFP-transduced cells using confocal microscopy, in particular, for colocalization of GCaMP6m-GFP + HuCD. We certainly plan to do further expression of GCaMP6m in vivo, but this was not the goal of this article.
- Clarifying how the authors determined the suitable temperature, incubation time, viral titer, kinds of tissues, kind of the viral serotype, despite the first attempt with rAAV to salmon? The selection of serotype, temperature, and incubation time would be particularly crucial.
In this work, we used a neuron-specific serotype AAV1 (AAV1.Camc2a.GCaMP6f.WPRE.bGHpA) for the mammalian hippocampus. The incubation time was determined as 1 week from the moment of rAAV transduction into the mesencephalic tegmentum. We defined this time interval as the minimum period after which an attempt was made to study the distribution of rAAV in the brain of juvenile salmon. This time is considered insufficient for the mammalian brain to obtain a sufficient number of transduced cells, in particular for in vivo studies. The temperature for keeping juvenile salmon after the rAAV injection in the aquarium was the usual 16C. Preliminary studies have shown that a shorter period after rAAV injection does not allow for a sufficient level of transduction in cells, in which case the rAAV-transduced cells were localized exclusively near the injection site.

Round 2
Reviewer 1 Report
All my comments addressed. Thanks.
Reviewer 2 Report
I have no more suggestions. I think it was revised according to previous reviews.